# Quantum Speedups for Sampling and Non-convex Optimization with Stochastic Zeroth Oracles

## Abstract

We propose quantum algorithms with provable speedups for sampling from probability distributions of the form $\pi \propto e^{-f}$, where $f : \mathbb{R}^d \mapsto \mathbb{R}$ is a potential function. In particular, we consider access only to a stochastic evaluation oracle, allowing simultaneous queries of the potential value at two different points under the same stochastic parameter. By introducing novel quantum algorithms for stochastic gradient estimation in this setting, our algorithms improve the evaluation complexities of classical samplers, such as Hamiltonian Monte Carlo (HMC) and Langevin Monte Carlo (LMC) in terms of dimension, precision, and other problem-dependent parameters. Furthermore, we demonstrate that our quantum sampling algorithms can be used to achieve quantum speedups in optimization, particularly for minimizing nonsmooth and approximately convex functions that commonly appear in empirical risk minimization problems.

## 1 Introduction

Efficient sampling from complex distributions is a central task across science and engineering, growing in importance as applications involve high-dimensional data and intricate probabilistic models. For example, in probabilistic machine learning, sampling facilitates posterior estimation and quantifies uncertainty in model predictions Welling & Teh (2011); Wang et al. (2015); Durmus & Moulines (2018); Roy et al. (2021). In non-convex optimization, sampling allows for the exploration of complex energy landscapes and helps avoid local minima Zhang et al. (2017); Chen et al. (2020). It is also widely used in other domains of science such as statistical mechanics Chandler (1987); Frenkel & Smit (2002), convex geometry Lovász & Vempala (2006); Cousins & Vempala (2018), etc.

Given a potential function $f : \mathbb{R}^d \to \mathbb{R}$, we consider the problem of sampling from a probability distribution $\pi$ of the form

$$\pi(\mathbf{x}) = \frac{e^{-f(\mathbf{x})}}{\int e^{-f(\mathbf{x})}\mathrm{d}\mathbf{x}}. \tag{1}$$

This distribution is called the *Gibbs-Boltzmann distribution*, and our goal is to efficiently sample approximately from $\pi$ while minimizing the number of evaluation queries in the *stochastic zeroth-order setting*. In this setting, we assume that we have only access to noisy function values $f(\mathbf{x}; \xi)$ where $\xi$ is a random variable that characterizes the noise.

Our first approach is based on discretization of continuous Langevin diffusion equation, which follows stochastic differential equation (SDE):

$$\mathrm{d}\mathbf{x}_t = -\nabla f(\mathbf{x}_t)\mathrm{d}t + \sqrt{2}\mathrm{d}\mathbf{B}(t), \tag{2}$$

where $\mathbf{B}(t)$ is the standard Brownian motion. The Euler-Maruyama discretization of this SDE results in the well-known *Langevin Monte Carlo* (LMC) algorithm:

$$\mathbf{x}_{t+1} = \mathbf{x}_t - \eta_t \nabla f(\mathbf{x}_t) + \sqrt{2\eta_t}\epsilon_t, \tag{3}$$

where $\eta_t > 0$ is the step size and $\epsilon_t$ is isotropic Gaussian noise. The second algorithm we consider is the *Hamiltonian Monte Carlo* (HMC) algorithm. HMC introduces the Hamiltonian $H(\mathbf{x}, \mathbf{p}) =$

$f(\mathbf{x}) + \frac{1}{2}\|\mathbf{p}\|^2$ and updates the position ($\mathbf{x}$) and momentum ($\mathbf{p}$) by simulating Hamiltonian dynamics, which follows the differential equations:

$$\frac{d\mathbf{x}}{dt} = \frac{\partial H}{\partial \mathbf{p}}, \quad \frac{d\mathbf{p}}{dt} = -\frac{\partial H}{\partial \mathbf{x}}. \tag{4}$$

Similar to LMC, in practice HMC is simulated by discretizing Eq. (4) and randomly refreshing the momentum periodically. We refer the reader to appendix A for more details on the discretization HMC algorithm. Despite their efficient convergence, the computational cost of each iteration in these algorithms becomes prohibitive when the computation of the gradient is not directly available. For example, in zeroth order setting one needs to compute the gradient using finite difference formula. Similarly, one uses mini-batch gradients when the objective function is given through large number of data points. To alleviate the computational burden, stochastic gradient-based samplers such as Stochastic Gradient Langevin Dynamics (SGLD) Welling & Teh (2011) and Stochastic Hamiltonian Monte Carlo (SG-HMC) Chen et al. (2014) have also been proposed. Instead of computing the gradient exactly, these algorithms use stochastic approximation to the gradient at each iteration. For example, the stochastic update for LMC becomes

$$\mathbf{x}_{t+1} = \mathbf{x}_t - \eta_t \mathbf{g}_t + \sqrt{2\eta_t}\epsilon_t. \tag{5}$$

where $\mathbf{g}$ is a random vectpr that approximates $\nabla f$. Although these algorithms originally proposed to address finite-sum settings, a stochastic gradient can also be obtained in zeroth order setting by using finite difference formulas by evaluating the function at two close points Nesterov & Spokoiny (2017). This technique is typically employed when autodifferentiation is costly due to expensive backpropagation in training ML models or the function value is only accessible in a black-box fashion. This scenario has been analyzed under various settings in optimization literature Duchi et al. (2015); Nesterov & Spokoiny (2017); Balasubramanian & Ghadimi (2022); Lin et al. (2022). For sampling problems, Roy et al. (2021) has analyzed the convergence of various discretizations of Langevin diffusion algorithms both for strongly convex and non-convex potentials using the noisy zeroth-order oracle. Similarly Dalalyan & Karagulyan (2019) has established the convergence of sampling under inexact gradients when the bias and the variance of the inexact estimates are bounded. Similarly, Yang & Wibisono (2023) analyzed the convergence of the inexact Langevin algorithm in KL divergence under different assumptions on the potential function.

Building on this classical foundation, researchers have started to explore how quantum computation could accelerate sampling and optimization in the past decade. Quantum algorithms such as multi-dimensional quantum mean estimation Cornelissen et al. (2022) and quantum gradient estimation Jordan (2005); Gilyén et al. (2019) have shown potential for reducing the query complexity of gradient-based methods van Apeldoorn et al. (2020); Chakrabarti et al. (2020); Sidford & Zhang (2023); Zhang et al. (2024); Liu et al. (2024). These techniques are particularly well-suited for addressing challenges in large-scale and noisy settings, as they can provide more accurate gradient estimates with asymptotically fewer queries. In this paper, we focus on integrating these quantum techniques to enhance the efficiency of stochastic gradient-based samplers and alleviate the computational burden inherent in classical methods.

The quest for quantum speedups for optimization has emerged since the discovery of Jordan's quantum gradient estimation algorithm Jordan (2005) and its refined version Gilyén et al. (2019) which have generated significant excitement by provable reductions in query complexity against classical finite-difference methods. Despite the effort in many years, the applicability of quantum gradient estimation is still rather limited in general settings of optimization problems, which is largely due to biased estimations, sensitivity to noise, or strong assumptions on the higher derivatives of the function. In this work, we bridge this technical gap by extending quantum gradient estimation to a more practical stochastic zeroth-order setting where the function values do not need to be accessed very precisely and the objective function does not need to satisfy impractical smoothness properties.

## 1.1 MAIN CONTRIBUTIONS

We summarize our contributions as follows.

- **Quantum Speedups for Gradient Estimation via Stochastic Evaluation Oracle**: We develop novel quantum gradient estimation algorithms under various smoothness assumptions in Section 2 to address the shortcomings of the existing gradient estimation methods. Our algorithm provides

quadratic speedup when the potential function is smooth, reducing the evaluation queries from $\tilde{\mathcal{O}}(\frac{d^2\sigma^2}{\epsilon^2})$ to $\tilde{\mathcal{O}}(\frac{d\sigma}{\epsilon})$ to compute the gradient up to $\epsilon$ accuracy (Theorem 2.4) where $\sigma^2$ is the variance of the noise as in Assumption 2.1. Furthermore, when the stochastic functions are also smooth with high probability, we manage to shave off an additional $d^{1/2}$ term (Theorem 2.7). This is achieved by combining quantum mean estimation with Jordan's quantum gradient estimation in a robust manner. Our gradient estimation algorithms could be useful as independent tools, especially in zeroth-order stochastic optimization.

- **Speedups for Zeroth-Order Sampling**: In Section 3, we combine our new quantum gradient estimation algorithm with HMC and LMC algorithms and analyzed the convergence of the final algorithm. We proved that our hybrid algorithms use fewer number of queries to evaluation oracle than the best known classical samplers under the same assumptions (Theorems 3.2 and 3.4).

- **Application to Non-Convex Optimization**: In Section 4, we extend our quantum sampling methods to optimize non-convex functions with specific structural properties, demonstrating that faster sampling translates to provable speedups in complex optimization tasks. In particular, we show that we can optimize non-smooth and approximately convex functions, i.e. a function that is uniformly close to a strongly convex function, using fewer stochastic evaluation queries than the best known classical algorithms in terms of dimension dependency (Theorem 4.5).

We note that the realization of the contributions in this paper requires a fault-tolerant quantum computer that can implement the procedures above with proper error correction. Therefore, our focus is on establishing the theoretical speedup as a guide for future algorithm design as empirical validation is not possible as of today.

## 1.2 Preliminaries

**Notation:** Bold symbols, such as $\mathbf{x}$ and $\mathbf{y}$, are used to represent vectors, with $\|\cdot\|$ indicating the Euclidean or operator norm depending on the context. Given two scalars $a$ and $b$, we use $a \wedge b$ to denote $\min\{a,b\}$ and use $a \vee b$ to denote $\max\{a,b\}$. We use $\mathcal{B}_d(c,r)$ to denote the $d$-dimensional ball centered at $c$ with radius $r$ and $G_d^l(c)$ to denote the $d$-dimensional grid centered at point $c$ with side length $l$. We occasionally use $G_d^l$ when the center of the grid is clear from the context. The notation $\tilde{O}$ is used to suppress the polylogarithmic dependencies on $d, \epsilon, L, \mu$ and $\alpha$ that will be defined later in the text.

**Quantum Computation:** Quantum computation is expressed using linear algebra over complex vector spaces. The computational basis of $\mathbb{C}^d$ is the standard basis $\{\mathbf{e}_0, \ldots, \mathbf{e}_{d-1}\}$, where $\mathbf{e}_i$ is the column vector with a 1 in the $(i+1)$st position and zeros elsewhere. In Dirac notation, we denote $\mathbf{e}_i$ by $|i\rangle$ and its conjugate transpose by $\langle i|$.

The state space of a quantum system with $n$ subsystems is the tensor product space $\mathbb{C}^{d_1} \otimes \cdots \otimes \mathbb{C}^{d_n}$. The tensor (Kronecker) product of two vectors $|u\rangle \in \mathbb{C}^{d_1}$ and $|v\rangle \in \mathbb{C}^{d_2}$ is the vector $|u\rangle \otimes |v\rangle \in \mathbb{C}^{d_1 d_2}$, given explicitly by: $|u\rangle \otimes |v\rangle = (u_0 v_0, u_0 v_1, \ldots, u_{d_1-1} v_{d_2-1})^\top$.

A single qubit is a normalized vector in $\mathbb{C}^2$, written as $\alpha|0\rangle + \beta|1\rangle$, with $|\alpha|^2 + |\beta|^2 = 1$. A system of $n$ qubits lives in the Hilbert space $\mathbb{C}^{2^n}$, and a general $n$-qubit state may be entangled, i.e., not expressible as a tensor product of single-qubit states. We often abbreviate tensor products such as $|u\rangle \otimes |v\rangle$ by $|u\rangle |v\rangle$.

Quantum operations correspond to unitary transformations. In the circuit model, a $k$-qubit gate is a unitary operator $U \in \mathbb{C}^{2^k \times 2^k}$. A universal gate set allows for any $n$-qubit unitary to be approximated using a sequence of two-qubit gates, up to arbitrarily small error. The gate complexity of a unitary operation refers to the number of such basic gates required in its circuit decomposition.

Measurement is the process of extracting classical information from a quantum system. A projective measurement of state $|\psi\rangle$ in the computational basis $\{|0\rangle, |1\rangle, \ldots, |2^n - 1\rangle\}$ yields outcome $i$ with probability $|\langle i|\psi\rangle|^2$, collapsing the state $|\psi\rangle$ to $|i\rangle$. In the quantum framework, a classical probability distribution $p$ can be represented by the quantum state $\sum_{\mathbf{x}} \sqrt{p(\mathbf{x})}|\mathbf{x}\rangle$. When measuring this state, the resulting outcomes are governed by the probability distribution $p$.

**Metrics:** We use several metrics to compare probability distributions over a state space $\mathcal{X}$. Let $\pi$ and $\mu$ be two probability distributions on $\mathcal{X}$. The $p$-Wasserstein distance between $\pi$ and $\mu$ is defined as $W_p(\pi, \mu) = \left(\inf_{\gamma \in \Gamma(\pi,\mu)} \mathbb{E}_{(\mathbf{x},\mathbf{y}) \sim \gamma} \|\mathbf{x} - \mathbf{y}\|^p\right)^{1/p}$ where $\Gamma(\pi, \mu)$ is the set of all joint distributions $\gamma(\mathbf{x}, \mathbf{y})$ whose marginals are $\pi$ and $\mu$. The KL divergence of $\pi$ with respect

to $\mu$ is defined as $\text{KL}(\pi\|\mu) = \int_{\mathcal{X}} \text{d}\mathbf{x}\pi(\mathbf{x}) \log\left(\frac{\pi(\mathbf{x})}{\mu(\mathbf{x})}\right)$ and the relative Fisher information is $\text{FI}(\pi\|\mu) = \int_{\mathcal{X}} \text{d}\mathbf{x}\pi(\mathbf{x}) \left\|\nabla \log\left(\frac{\pi(\mathbf{x})}{\mu(\mathbf{x})}\right)\right\|^2$. The total variation distance is defined as $\text{TV}(\pi,\mu) = \sup_{A\subseteq\mathcal{X}} |\pi(A) - \mu(A)| = \frac{1}{2}\int_{\mathcal{X}} \text{d}\mathbf{x}|\pi(\mathbf{x}) - \mu(\mathbf{x})|$.

## 2 QUANTUM GRADIENT ESTIMATION IN ZEROTH-ORDER STOCHASTIC SETTING

We consider access to an evaluation oracle for the stochastic components $f_\xi(\mathbf{x}) = f(\mathbf{x};\xi)$, where $\xi \in \Xi$ represents a random seed characterizing the randomness of the noise. Then, the function is accessed through the binary oracle below.

$$O_f \ket{\mathbf{x}} \ket{\xi} \ket{0} \mapsto \ket{\mathbf{x}} \ket{\xi} \ket{f(\mathbf{x};\xi)}. \tag{6}$$

We characterize the complexity of our algorithms in this section with respect to this oracle. Jordan's quantum gradient estimation algorithm Jordan (2005) constructs a superposition over $d$ dimensional $N$ grid points $G_d^l$ centered around the point $\mathbf{x}$ with side length $l$. Then the following quantum state is obtained by using a phase oracle $O : \ket{\mathbf{x}} \mapsto e^{itf(\mathbf{x})}\ket{\mathbf{x}}$,

$$\ket{\psi} = \frac{1}{\sqrt{N^d}} \sum_{\mathbf{y}\in G_d^l(\mathbf{x})} e^{it[f(\mathbf{x}+\frac{l}{N}(\mathbf{y}+\frac{l}{N})-f(\mathbf{x})]} \ket{\mathbf{y}} \tag{7}$$

where $t$ is a proper scale factor. Then the algorithm uses quantum Fourier transform to compute the full gradient. This algorithm uses only constant number of queries to the function (Lemma A.4) whereas one needs at least $d$ queries to approximate the gradient classically. We refer the reader to A.4 for more detailed review on Jordan's quantum estimation algorithm. We occasionally refer to the version of this algorithm with optimized parameters as `QuantumGradient` given in pseudocode 4. Although Jordan's algorithm is appealing as it only uses a constant number of evaluations to estimate the gradient, its practical use cases remained limited as it requires the function evaluations to be very accurate and the function needs to be very close to linear. In fact when the function is not linear, the speedup in dimension does not always survive Gilyén et al. (2019). In this section, we develop algorithmic techniques to address these obstacles.

### 2.1 QUANTUM GRADIENT ESTIMATION FOR SMOOTH POTENTIALS

We assume the following about the potential function.

**Assumption 2.1** (Bounded Noise). For any $\mathbf{x} \in \mathbb{R}^d$, the stochastic zeroth-order oracle outputs an estimator $f(\mathbf{x};\xi)$ of $f(\mathbf{x})$ such that $\mathbb{E}[f(\mathbf{x};\xi)] = f(\mathbf{x})$, $\mathbb{E}[\nabla f(\mathbf{x};\xi)] = \nabla f(\mathbf{x})$, and $\mathbb{E}\|\nabla f(\mathbf{x};\xi) - \nabla f(\mathbf{x})\|^2 \leq \sigma^2$.

**Assumption 2.2** (Smoothness). The potential function $f : \mathbb{R}^d \to \mathbb{R}$ has $L$-Lipschitz gradients. Specifically, it holds that

$$\|\nabla f(\mathbf{x}) - \nabla f(\mathbf{y})\| \leq L\|\mathbf{x} - \mathbf{y}\|.$$

These assumptions are standard in the zeroth-order sampling and optimization literature Roy et al. (2021); Balasubramanian & Ghadimi (2022). We note that Assumption 2.1 is broader than an additive noise model, as it accommodates models with multiplicative noise. For example, suppose that $f : \mathcal{B}_d(0,R) \mapsto \mathbb{R}$ is an $L$ smooth differentiable function, and that the stochastic components are of the form $f(\mathbf{x};\xi) = \xi f(\mathbf{x})$, where $\mathbb{E}[\xi] = 1$ and $\mathbb{E}[\xi^2] \leq \frac{\sigma^2}{4L^2R^2}$. In this case, Assumption 2.1 is satisfied.

Suppose that the function $f$ can be queried with the same randomness at two different points, that is, we can query $f(\mathbf{x};\xi_i)$ and $f(\mathbf{y};\xi_i)$ simultaneously [1]. For example, in finite-sum case, one can first prepare the quantum state $\frac{1}{\sqrt{n}}\sum_{i=1}^n \ket{0}\ket{i}$ and then prepare the quantum state $\frac{1}{\sqrt{n}}\sum_{i=1}^n \ket{f(\mathbf{x};i)}\ket{i}$ using binary oracle, since $f$ can be queried at each data point and superposition is over indices 1 to $n$. Note that this does not require any data encoding (or QRAM structure) as superposition is just over the indices. Classically, the gradient in this two-point setting can be estimated using the Gaussian

---

[1]This is the case in finite-sum and some bandit settings where $\xi$ can be queried explicitly.

smoothing technique. This involves sampling random directions from the extended space around the target point and performing two-point evaluations to approximate the gradient. Specifically, the gradient can be approximated as:

$$\mathbf{g}_{\nu,b}(\mathbf{x}) = \frac{1}{b} \sum_{i=1}^{b} \frac{f(\mathbf{x} + \nu \mathbf{u}_i; \xi_i) - f(\mathbf{x}; \xi_i)}{\nu} \mathbf{u}_i, \tag{8}$$

where $\mathbf{u}_i \sim \mathcal{N}(0, I_d)$ are independent and identically distributed random vectors. Balasubramanian & Ghadimi (2022) showed that for any $\mathbf{x} \in \mathbb{R}^d$, the estimator $\mathbf{g}_{\nu,b}$ satisfies $\mathbb{E}\|\mathbf{g}_{\nu,b}(\mathbf{x}) - \nabla f(\mathbf{x})\|^2 \leq \frac{4(d+5)(\|\nabla f(\mathbf{x})\|^2 + \sigma^2)}{b} + \frac{3\nu^2 L^2 (d+3)^3}{2}$. Although the squared norm of the gradient on the right-hand side is unbounded, it is typically of order $\tilde{\mathcal{O}}(d)$ in expectation throughout the trajectory of LMC (See Eq. (60)). Consequently, this method requires $b = \mathcal{O}(d^2/\epsilon^2)$ function evaluations to achieve an $\epsilon$-accurate gradient estimate in the $L_2$ norm.

As far as we are aware of, currently there does not exist a quantum algorithm to estimate the gradient in this particular setting. Our purpose to design additional techniques to be able to use Jordan's algorithm in this setting. The major technical tool in the algorithm is the phase oracle in the following proposition.

**Proposition 2.3.** *Let $X \in \mathbb{R}$ be a random variable such that $\mathbb{E}\|X - \mathbb{E}[X]\|^2 \leq \sigma^2$. Given two reals $t \geq 0$ and $\epsilon \in (0, 1)$, then there is a unitary operator $P_{t,\epsilon}^X : |0\rangle |0\rangle \mapsto |\phi_X\rangle |0\rangle$ acting on $\mathcal{H}_X \otimes \mathcal{H}_{aux}$ that can be implemented using $\tilde{\mathcal{O}}(t\sigma \log(1/\epsilon))$ quantum experiments and binary oracle queries to $X$ such that*

$$\| |\phi_X\rangle - e^{it\mathbb{E}[X]} |0\rangle \| \leq \epsilon,$$

*with probability at least $8/9$.*

This phase oracle is similar to the oracle implemented in Cornelissen et al. (2022); however, their algorithm requires $\|X\| \leq 1$ whereas $\|X\|$ might be unbounded in our case. Hence, Proposition 2.3 generalizes the phase oracle to the unbounded random variables by constructing a sequence of unitaries for different levels of truncation of the random variable $X$ (See the more detailed description in Appendix D). In particular, this oracle replaces phase oracle (line 4 in Algorithm 4) where the random variable is the stochastic function $f(x; \xi)$ in this case. Since Jordan's algorithm is biased and succeeds with high probability, we can also obtained smooth and unbiased gradient by postprocessing the output using the Multi-Level Monte Carlo technique (Algorithm 3). The preliminaries for the MLMC algorithm can be found in Appendix A.2.

**Theorem 2.4.** *Suppose that the potential function $f$ satisfies Assumptions 2.1 and 2.2 and further suppose that $\|\nabla f(\mathbf{x})\| \leq M^2$ for all $\mathbf{x}$. Then, given a real $\hat{\sigma} > 0$, there exists a quantum algorithm that outputs a random vector $\mathbf{g}$ such that*

$$\mathbb{E}[\mathbf{g}] = \nabla f(\mathbf{x}), \quad and \quad \mathbb{E}\|\mathbf{g} - \nabla f(\mathbf{x})\|^2 \leq \hat{\sigma}^2$$

*using $\tilde{O}(\frac{\sigma d}{\hat{\sigma}})$ queries to the stochastic evaluation oracle.*

Proofs of Proposition 2.3 and Theorem 2.4 are postponed to Appendix D due to limited state.

Although 2.4 is motivated by the sampling task, it might be of an independent tool for various optimization problems even in nonsmooth setting. For example, suppose that $f_\xi$ is a non-smooth but locally $L$-Lipschitz function around the grid $G_d^l$. We define $f_v(\mathbf{x}) = \mathbb{E}_{\xi \in \Xi, \mathbf{u} \sim \mathcal{B}(0,1)}[f(\mathbf{x} + v\mathbf{u}; \xi)]$. Then, let $\mathbf{y} \in G_d^l$, $\mathbb{E}\|\nabla f(\mathbf{y} + v\mathbf{u}) - \nabla f_v(\mathbf{y})\|^2 \leq 4L^2$. It is known that $f_v$ is a smooth function with smoothness parameter $O(Ld^{1/2}v^{-1})$. Hence, by Theorem 2.4 our algorithm outputs an unbiased estimator $\mathbf{g}$ such that $\mathbb{E}[\mathbf{g}] = \nabla f_v(\mathbf{x})$ and $\mathbb{E}\|\mathbf{g} - \nabla f_v(\mathbf{x})\|^2 \leq \hat{\sigma}^2$ using $\tilde{\mathcal{O}}(\frac{Ld}{\hat{\sigma}})$ queries to $f_\xi$. This result has recently been established in Liu et al. (2024), and it is a special case of Theorem 2.4. Hence, our quantum gradient estimation can give speedups beyond the settings considered in Liu et al. (2024) but this is outside of the scope of this work.

---

[2] One can show that the norm of the gradient is bounded by a function of problem parameters throughout the trajectory of HMC or LMC due to smoothness. Since the dependency on $M$ is logarithmic, we do not give an explicit bound on $M$.

## 2.2 Quantum Gradient Estimation under Additional Smoothness Assumption

In this section, we consider a setting that imposes a slightly stronger smoothness assumption on the stochastic functions $f_\xi$ to be able to improve the dimension dependency further.

**Assumption 2.5** (Lipschitz Stochastic Gradients)**.** The stochastic component $f(\cdot; \xi) : \mathbb{R}^d \to \mathbb{R}$ has $L(\xi)$-Lipschitz gradients for any $\xi \in \Xi$. Specifically, it holds that

$$\|\nabla f(\mathbf{x}; \xi) - \nabla f(\mathbf{y}; \xi)\| \le L(\xi)\|\mathbf{x} - \mathbf{y}\|, \tag{9}$$

and the expected Lipschitz constant satisfies $\mathbb{E}[L(\xi)] = L$.

Assumption 2.5 is weaker than the assumption that each stochastic function $f_\xi$ has $L$-Lipschitz gradients and it is straightforward to show that Assumption 2.5 implies that $f$ has Lipschitz gradients.

In this section, we assume the following sampling oracle,

$$O_\xi : |\mathbf{x}\rangle \mapsto \sum_{\xi \in \Xi} \sqrt{\Pr(\xi)} \, |\mathbf{x}\rangle \, |\xi\rangle \,. \tag{10}$$

As opposed to implementing an accurate phase oracle, one can estimate the gradient $\nabla f(\mathbf{x}; \xi)$ and then use the quantum mean estimation algorithm to compute $\nabla f(\mathbf{x})$. Quantum mean estimation is a technique to compute the mean of a random variable quadratically faster than classical techniques (See A.3 overview of quantum mean estimation).

However, Assumption 2.5 implies that $f_\xi$ might not be a smooth function (even if $f$ is smooth), which is the requirement in Lemma A.4. Hence, Jordan's algorithm might fail to compute the gradient for $\nabla f_\xi$ with small probability no matter how large we set $\beta$ in Algorithm 4. To address this, we propose a robust version of the quantum gradient estimation algorithm. Our final algorithm achieves $\tilde{\mathcal{O}}(d^{1/2}\epsilon^{-1})$ query complexity to estimate the gradient up to $\epsilon$ error.

---

**Algorithm 1** `QuantumStochasticGradient`

---

0: **Input**: stochastic functions $f_\Xi$, variance $\sigma^2$, target $\epsilon$, smoothness parameter $L$, point $\mathbf{x}$.
   Define $\beta = \frac{164 L \sigma^2}{\epsilon^2}$, $D = \frac{40\sigma^2}{\epsilon}$, $\epsilon' = \frac{\epsilon^2}{\beta^2 d^3 (12000)^2}$.
1: Sample $\xi_0$ at random from $\Xi$.
2: Compute $\boldsymbol{s} = $ `QuantumGradient`$(f_{\xi_0}, \epsilon', M, \beta, \mathbf{x})$ (See algorithm 4).
3: Let $\mathcal{A}$ be a randomized algorithm that runs $\mathbf{g} = $ `QuantumGradient`$(f_\xi, \epsilon', M, \beta, \mathbf{x})$ with random $\xi \in \Xi$ and outputs $\mathbf{g}$ if $\|\mathbf{g} - \boldsymbol{s}\| \le D$, otherwise it outputs $\boldsymbol{s}$. Further suppose that $\mathcal{A}$ does not make any measurement.
4: Run $\mathcal{A}$ on the superposition state to obtain $\sum_{\xi \in \Xi} \sqrt{\Pr(\xi)} \, |\mathcal{A}(\mathbf{x})\rangle \, |\xi\rangle \,.$
5: Output $\boldsymbol{v} = $ `QuantumMeanEstimation`$(\mathcal{A}, \epsilon/4, \delta)$.

---

The algorithm (line 2) first selects a random $\xi$ and computes runs the gradient estimation algorithm and stores the output $\mathbf{s}$. The crucial idea is to use this value to replace very bad estimates and then show that this procedure does not increase the error with high probability. Next, we run the algorithm $\mathcal{A}$ in superposition on $|\psi_1\rangle = \sum_{\xi \in \Xi} \sqrt{\Pr(\xi)} \, |\mathbf{x}\rangle \, |\xi\rangle$. Then the last step in Jordan's algorithm is the following quantum state,

$$\sum_{\xi \in \Xi} \sqrt{\Pr(\xi)} \, |\tilde{\mathbf{g}}(\mathbf{x}; \xi)\rangle \, |\mathbf{x}\rangle \, |\xi\rangle + |\mathcal{X}_1\rangle \,, \tag{11}$$

where $|\mathcal{X}_1\rangle$ is another garbage state with a small amplitude and

$$\tilde{\mathbf{g}}(\mathbf{x}, \xi) = \begin{cases} \mathbf{g}(\mathbf{x}, \xi) & \text{if } \|\mathbf{g}(\mathbf{x}, \xi) - \mathbf{s}\| \le D, \\ \mathbf{s} & \text{otherwise.} \end{cases} \tag{12}$$

is the corrected gradient estimate.

Finally, we estimate the mean of the first register to compute $\boldsymbol{v}$, which is output as the gradient estimate. Note that the reason why replacement with $\tilde{\mathbf{g}}$ works is the fact that the mediocre estimate $\mathbf{s}$ is one standard deviation away from the true gradient with high probability. Therefore, replacing the

tails of the distribution (the estimates with very high error) actually does not change the expectation too much. However, this procedure is necessary as very erroneous values can change the expectation without replacement. A more detailed analysis by more rigorous probabilistic arguments gives the following results.

**Lemma 2.6.** *Under Assumptions 2.1 and 2.5, Algorithm 1 returns a vector $\boldsymbol{v}$ such that*

$$\|\boldsymbol{v} - \nabla f(\mathbf{x})\| \leq \epsilon \tag{13}$$

*with high probability using $\tilde{\mathcal{O}}(\sigma d^{1/2}\epsilon^{-1})$ queries to the stochastic evaluation oracle.*

Next, we can postprocess the output of Algorithm 1 using MLMC to obtain a smooth and unbiased estimate.

**Theorem 2.7** (Smooth Gradient). *Suppose that the potential function $f$ satisfies Assumptions 2.1 and 2.5 and further suppose that $\|\nabla f(\mathbf{x})\| \leq M$ for all $\mathbf{x}$. Then, given a real $\hat{\sigma} > 0$, there exists a quantum algorithm that outputs a random vector $\mathbf{g}$ such that*

$$\mathbb{E}[\mathbf{g}] = \nabla f(\mathbf{x}), \quad \text{and} \quad \mathbb{E}\|\mathbf{g} - \nabla f(\mathbf{x})\|^2 \leq \hat{\sigma}^2 \tag{14}$$

*using $\tilde{\mathcal{O}}(\frac{\sigma d^{1/2}}{\hat{\sigma}})$ queries to the stochastic evaluation oracle in expectation.*

The mathematical details of Lemma 2.6 and Theorem 2.7 are postponed to Appendix D. We remark our focus is on the query complexity and it can be shown that quantum mean estimation and Jordan's algorithm use $\mathcal{O}(\text{poly}(d, \log(1/\epsilon)))$ gates Cornelissen et al. (2022); Chakrabarti et al. (2025) and our algorithm does not change the circuit structure.

# 3 QUANTUM SPEEDUPS FOR SAMPLING VIA ZEROTH-ORDER ORACLE

We apply our quantum gradient estimation algorithm to establish the convergence of both HMC and LMC in strongly convex and LSI settings, respectively. In particular, at each iteration, we use the inexact gradients computed by our quantum gradient estimation algorithms introduced in previous sections. Our proof techniques involve establishing the convergence rates of HMC and LMC with inexact gradients. As large error in the gradient requires more iterations and smaller error requires more function evaluations by the gradient estimation algorithm, we optimize the error in the gradient to obtain the total optimal cost. We first present our results for approximate sampling from strongly logconcave distributions using HMC algorithm where the error metric is Wasserstein metric.

**Assumption 3.1** (Strong Convexity). *There exists a positive constant $\mu$ such that for all $\mathbf{x}, \mathbf{y} \in \mathbb{R}^d$ it holds that*

$$f(\mathbf{x}) \geq f(\mathbf{y}) + \langle \nabla f(\mathbf{y}), \mathbf{y} - \mathbf{x} \rangle + \frac{\mu}{2}\|\mathbf{x} - \mathbf{y}\|^2. \tag{15}$$

We define the *condition number* $\kappa := \frac{L}{\mu}$.

**Theorem 3.2** (Main Theorem for `QZ-HMC`). *Let $\mu_k$ be the distribution of $\mathbf{x}_k$ in `QZ-HMC` algorithm. Suppose that $f$ satisfies Assumption 3.1. Given that the initial point $\mathbf{x}_0$ satisfies $\|\mathbf{x}_0 - \arg\min_{\mathbf{x}} f(\mathbf{x})\| \leq \frac{d}{\mu}$, if we set the step size $\eta = \mathcal{O}\left(\frac{\epsilon}{d^{1/2}\kappa^{3/2}}\right)$, $S = \tilde{\mathcal{O}}\left(\frac{Ld^{1/2}\kappa^{3/2}}{\epsilon}\right)$, $T = \tilde{\mathcal{O}}(1)$, and $\hat{\sigma}^2 = \mathcal{O}\left(\frac{L^{3/2}d^{1/2}\epsilon}{\kappa^{3/2}}\right)$, we have*

$$W_2(\mu_{ST}, \pi) \leq \epsilon.$$

*In addition, under Assumptions 2.1 and 2.2, the query complexity to the stochastic evaluation oracle is $\tilde{\mathcal{O}}\left(\frac{d^{5/4}\sigma}{\epsilon^{3/2}}\right)$ or under Assumptions 2.1 and 2.5 the query complexity to the stochastic evaluation oracle is $\tilde{\mathcal{O}}\left(\frac{d^{3/4}\sigma}{\epsilon^{3/2}}\right)$.*

We note that if the initial point does not satisfy the closeness condition, it can be obtained using $\mathcal{O}(1)$ iterations of SGD Baker et al. (2019). The proof of this theorem is postponed to Appendix B.1. The closest result in the classical setting is given by Roy et al. (2021) for Kinetic LMC algorithm which is obtained by setting the inner iterations to 1 in HMC algorithm. Their classical evaluation complexity under Assumptions 2.1 and 2.2 is $\tilde{\mathcal{O}}(d^2\sigma^2/\epsilon^2)$ for convergence in $W_2$ distance (Theorem 2.2 in Roy

et al. (2021)). Our algorithm uses $\tilde{\mathcal{O}}(d^{5/4}\sigma/\epsilon^{3/2})$ evaluation queries providing speedup both in $d$, $\epsilon$, and $\sigma$. It is worth noting that quantum acceleration of sampling in the zeroth-order setting for strongly convex functions has been studied in Childs et al. (2022) when the error in the function evaluation is $\epsilon$; however, their results do not apply to bounded variance setting. Similarly the results by Dalalyan & Karagulyan (2019); Yang & Wibisono (2023) either use different assumptions on $f$ or the access model.

As strong convexity and Wasserstein metric might be restrictive, we consider the sampling problem under for nonconvex potential functions. We assume that the target distribution satisfies the log-Sobolev inequality.

**Assumption 3.3** (Log-Sobolev Inequality). We say that $\pi$ satisfies the Log-Sobolev inequality with constant $\alpha$ if for all $\rho$, it holds that

$$\mathrm{KL}(\rho||\pi) \leq \frac{1}{2\alpha}\mathrm{FI}(\rho||\pi). \tag{16}$$

This is a sampling analog of the PL (Polyak-Łojasiewicz) condition commonly used in optimization Chewi & Stromme (2024) and standard in non-log-concave sampling literature Vempala & Wibisono (2019); Ma et al. (2019); Chewi et al. (2022); Kinoshita & Suzuki (2022). We note that LSI relaxes strong convexity in the sense that for any $\mu$ strongly convex function $f$, $\pi$ satisfies the Log-Sobolev inequality with constant $\frac{\mu}{2}$. This assumption is weaker than the dissipative gradient condition Raginsky et al. (2017); Zou et al. (2019) which is used commonly in non-log-concave sampling. We first present the main result and defer the proof to the appendix.

**Theorem 3.4** (Main Theorem for QZ-LMC). *Under Assumption 3.3, let $\mu_k$ be the distribution of $\mathbf{x}_k$ in QZ-LMC algorithm. Then, if we set the step size $\eta = \mathcal{O}\left(\frac{\epsilon\alpha}{dL^2}\right)$, $K = \tilde{\mathcal{O}}\left(\frac{dL^2 \log(\mathrm{KL}(\mu_0||\pi))}{\epsilon\alpha^2}\right)$, and $\hat{\sigma}^2 = \mathcal{O}(\alpha\epsilon)$, we have*

$$\left\{\mathrm{KL}(\mu_K||\pi), \mathrm{TV}(\mu_K, \pi)^2, \frac{\alpha}{2}\mathrm{W}_2(\mu_K, \pi)^2\right\} \leq \epsilon.$$

*In addition, under Assumptions 2.1 and 2.2, the query complexity to the stochastic evaluation oracle is $\tilde{\mathcal{O}}\left(\frac{d^2 L^2\sigma}{\alpha^{5/2}\epsilon^{3/2}}\right)$, or under Assumptions 2.1 and 2.5 the query complexity to the stochastic evaluation oracle is $\tilde{\mathcal{O}}\left(\frac{d^{3/2}L^2\sigma}{\alpha^{5/2}\epsilon^{3/2}}\right)$.*

Note that this result holds for various distance metrics. Comparing to the classical results, Roy et al. (2021) analyzed the convergence of LMC in the zeroth-order setting under Assumptions 2.1 and 2.2 and established evaluation complexity $\mathcal{O}(d^3\sigma^2/\epsilon^4)$ for convergence in $\mathrm{W}_2$ distance (Theorem 3.2 in Roy et al. (2021)). Our algorithm uses $\tilde{\mathcal{O}}(d^2\sigma/\epsilon^3)$ evaluation queries under the same assumptions giving polynomial speedups in $d, \sigma$ and $\epsilon$.

## 4 APPLICATION TO NONSMOOTH AND NONCONVEX OPTIMIZATION

Although we assumed smoothness for $f$ in previous sections, our results also help in nonsmooth sampling and optimization problems. In this section, we consider the following problem. Suppose that $f$ satisfies the following assumptions.

**Assumption 4.1** (Approximate-Convexity). Let $f$ be a differentiable function, we say that $f : \mathbb{R}^d \to \mathbb{R}$ is an $\epsilon$-approximately convex function, if there exists a strongly convex function $F$ such that for all $\mathbf{x}$,

$$|F(\mathbf{x}) - f(\mathbf{x})| \leq \frac{\epsilon}{d}. \tag{17}$$

Instead of smoothness, we only assume that $f$ is Lipschitz continuous.

**Assumption 4.2.** For all $\mathbf{x}, \mathbf{y} \in \mathbb{R}$, $f : \mathbb{R}^d \to \mathbb{R}$ satisfies,

$$|f(\mathbf{x}) - f(\mathbf{y})| \leq M\|\mathbf{x} - \mathbf{y}\|. \tag{18}$$

The goal is to find an approximate minimizer $\mathbf{x}^\star$ such that $|f(\mathbf{x}^\star) - \min_{\mathbf{x}} f(\mathbf{x})| \leq \epsilon$. Although these assumptions are very strong, this problem commonly appears in empirical risk minimization

(ERM) tasks where $f$ is given by empirical observations. Even the population function is smooth and strongly convex, the empirical optimization function loses its smoothness and convexity due to subsampling. Similar settings have also appeared in the context of escaping from local minima both in classical Belloni et al. (2015) and quantum settings Li & Zhang (2024) with access to a stochastic evaluation oracle. Diverging from previous techniques, we address these problems by sampling from Gibbs sampling by using smooth approximation of $f$ as proxy. Since $f$ is not smooth, we consider the smoothed approximation

$$f_v(\mathbf{x}) = \mathbb{E}_{u \sim \mathcal{B}_d(0,1)}[f(\mathbf{x} + v\mathbf{u})] \tag{19}$$

where $\mathcal{B}$ is unit $L_2$ ball around the center. The local properties of $f_v$ are known and given by the following proposition.

**Proposition 4.3.** *If $f$ satisfies Assumption 4.2, then $f_v$ satisfies*

- $|f_v(\cdot) - f(\cdot)| \le vM$ *and* $|f_v(\mathbf{x}) - f_v(\mathbf{y})| \le L\|\mathbf{x} - \mathbf{y}\|,$

- $|\nabla f(\mathbf{x}) - \nabla f(\mathbf{y})| \le cM\sqrt{d}v^{-1}$ *for some constant $c > 0$.*

First we notice that,

$$\mathbb{E}_{\mathbf{u}}\|\nabla f(\mathbf{x} + v\mathbf{u}) - \nabla f_v(\mathbf{x})\|^2 \le 4M^2 \tag{20}$$

as $\|\nabla f(x)\| \le M$ because of Lipschitz continuity. Hence, Assumption 2.1 holds with $\sigma^2 = 4M^2$ and Assumption 2.2 holds with $L = \frac{cM\sqrt{d}}{v}$. Therefore, using Theorem 3.4, we can sample from the Gibbs-Boltzmann distribution with potential $f_v$ faster than the classical algorithms. This is possible due to the fact that our quantum gradient estimation can compute the gradient of $f_v$ if we consider $f(x + v\mathbf{u})$ as the stochastic evaluation oracle. Then the seed set $\Xi = \mathcal{B}(0, 1)$ for which the sampling oracle can be prepared very efficiently. Since initial goal is to optimize $f$ rather than to sample from the Gibbs distribution, we use the following lemma that describes a method to turn a sampling algorithm into an optimizer.

**Lemma 4.4.** *Let $\pi_v^\beta = \frac{e^{-\beta f_v(\mathbf{x})}}{\int e^{-\beta f_v(\mathbf{x})}d\mathbf{x}}$. If $\beta = \Theta(d/\epsilon)$ and $v \le \mathcal{O}(\frac{\epsilon}{Md})$, then sampling from $\pi_v^\beta$ returns $\epsilon$ approximate optimizer for $f$ with probability at least 0.1.*

The essential idea is that Gibbs distribution gets localized around the global minima of $f$ as $\beta$ increases. One other challenge to sample from $f_v$ is that the Log-Sobolev constant in general might depend on dimension. However, by using Holley–Strook LSI perturbation result, we showed the explicit dimension dependence of $\alpha$ to characterize the total evaluation complexity in E. Next, we give our main result.

**Theorem 4.5.** *Suppose that $f$ satisfies Assumptions 4.1 and 4.2. Then, there exists a quantum algorithm that returns $\epsilon$ approximate minimizer for $f$ with probability at least 0.1 using $\tilde{\mathcal{O}}\left(\frac{d^{9/2}}{\epsilon^{3/2}}\right)$ queries to the stochastic evaluation oracle for $f$.*

The proof of Lemma 4.4 and Theorem 4.5 are postponed to Appendix E. The closest result to our setting is given by Li & Zhang (2024) and their query complexity in the stochastic setting is $\tilde{\mathcal{O}}(d^5/\epsilon)$ although their assumptions are slightly different. First, they assume that the noise is sub-Gaussian and additive. Furthermore, they assume $F$ is convex in a bounded domain but not necessarily strongly convex. Noting that these differences might possibly make the classical results loose, our algorithm seems to give a speedup in dimension dependence with a small performance drop in terms of $\epsilon$. However, this is a known trade-off in sampling algorithms. Since their algorithm uses a reversible sampler (hit-and-run walk), their $\epsilon$ dependence only comes from the quantum mean estimation. On the other hand, our algorithm uses a non-reversible sampler (also referred to as a low accuracy sampler) which typically gives better dependency on dimension but worse on accuracy. We also note that the classical algorithm by Belloni et al. (2015) takes $\tilde{\mathcal{O}}(\frac{d^{7.5}}{\epsilon^2})$ queries to the stochastic evaluation oracle.

Upon completion of this work, we became aware of recent studies by Augustino et al.Augustino et al. (2025) and Chakrabarti et al.Chakrabarti et al. (2025), which also investigate zeroth-order stochastic convex optimization under assumptions similar to those in Li & Zhang (2024). They propose algorithms with query complexities of $\widetilde{\mathcal{O}}(d^{9/2}/\epsilon^7)$ and $\widetilde{\mathcal{O}}(d^3/\epsilon^5)$, respectively. While both approaches exhibit worse dependence on $\epsilon$ compared to ours, we emphasize that the assumptions and problem settings differ significantly than ours.

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

## A   BACKGROUND

In this section, we give background on some classical and quantum algorithms for various tasks that are repeatedly referred in the main text.

### A.1   OVERVIEW OF HAMILTONIAN MONTE CARLO ALGORITHM

Hamiltonian Monte Carlo (HMC) is an advanced sampling technique designed to efficiently explore high-dimensional probability distributions by introducing auxiliary momentum variables. Given a target distribution $\pi(\mathbf{x}) \propto e^{-f(\mathbf{x})}$, HMC augments the state space with momentum variables $\mathbf{p}$ and defines the Hamiltonian $H(\mathbf{x}, \mathbf{p}) = f(\mathbf{x}) + \frac{1}{2}\|\mathbf{p}\|^2$ where $\mathbf{p} \sim \mathcal{N}(0, I)$.

HMC alternates between updating the position $\mathbf{x}$ and momentum $\mathbf{p}$ by simulating Hamiltonian dynamics Eq. (4). In practice, Hamiltonian dynamics is simulated using the leapfrog integrator, which discretizes the continuous equations of motion. The key advantage of HMC is that it allows for large, efficient moves through the parameter space by leveraging gradient information and auxiliary momentum. This reduces the correlation between successive samples, particularly in high-dimensional spaces, resulting in faster convergence compared to simple random-walk methods like the Metropolis-Hastings algorithm. In practice, Hamiltonian dynamics are simulated using the leapfrog integrator, which discretizes the continuous equations of motion. The leapfrog method proceeds in three steps:

$$\mathbf{p}_{k+\frac{1}{2}} = \mathbf{p}_k - \frac{\eta}{2}\nabla f(\mathbf{x}_k),$$

$$\mathbf{x}_{k+1} = \mathbf{x}_k + \eta\mathbf{p}_{k+\frac{1}{2}},$$

$$\mathbf{p}_{k+1} = \mathbf{p}_{k+\frac{1}{2}} - \frac{\eta}{2}\nabla f(\mathbf{x}_{k+1}),$$

where $\eta$ is the step size. After a series of updates, the momentum $\mathbf{p}_{k+1}$ is refreshed by sampling from $\mathcal{N}(0, I)$. This discretization ensures symplecticity, preserving volume in phase space and allowing the algorithm to make large, energy-conserving moves through the parameter space.

---

**Algorithm 2** `SG-HMC`

---

**input** The stochastic gradient oracle $O_{\nabla f}$, initial point $\mathbf{x}_0$, step size $\eta$, number of leapfrog steps $S$, number of HMC proposals $T$

**output** Approximate sample from $\pi \propto e^{-f(\mathbf{x})}$

  **for** $t = 0$ to $T$ **do**

    Sample $\mathbf{p}_{St} \sim \mathcal{N}(0, I)$

    **for** $s = 0$ to $S - 1$ **do**

      $k = St + s$

      $\mathbf{x}_{k+1} = \mathbf{x}_k + \eta\mathbf{p}_k - \frac{\eta^2}{2}\mathbf{g}(\mathbf{x}_k, \boldsymbol{\xi}_k)$

      $\mathbf{p}_{k+1} = \mathbf{p}_k - \frac{\eta}{2}\mathbf{g}(\mathbf{x}_k, \boldsymbol{\xi}_k) - \frac{\eta}{2}\mathbf{g}(\mathbf{x}_{k+1}, \boldsymbol{\xi}_{k+1/2})$

    **end for**

  **end for**

  **Return** $\mathbf{x}^T$

---

Similar to SGLD, one can replace the gradients with stochastic gradients resulting in SG-HMC (See Algorithm 2). The stochastic gradients $\mathbf{g}(\mathbf{x}, \xi)$ in Algorithm 2 can be obtained using different techniques such as mini-batch, SVRG, CV, or even zeroth-order methods. In this case, we use quantum variance reduction techniques to compute $\mathbf{g}(\mathbf{x}, \xi)$.

### A.2   OVERVIEW OF MULTI-LEVEL MONTE CARLO ALGORITHM

In this section, we give a brief overview of a technique known as the Multi-Level Monte Carlo algorithm. Without using this technique, our gradient estimation algorithms would not provide an unbiased estimate for the gradient. Suppose that we have an algorithm `BiasedStochasticGradient`$(\mathbf{x}, \sigma)$ that outputs $\mathbf{v}$ such that $\mathbb{E}\|\mathbf{v} - \nabla f(\mathbf{x})\| \leq \hat{\sigma}^2$ with cost $\tilde{\mathcal{O}}\left(\frac{C}{\sigma}\right)$ where $C$ is a function of other problem parameters. Consider the following algorithm.

---

**Algorithm 3** `UnbiasedStochasticGradient`

---

0: **Input**: Estimator `BiasedStochasticGradient`, target variance $\hat{\sigma}^2$
   **Output**: An unbiased estimate $\mathbf{g}$ of $\nabla f(\mathbf{x})$ with variance at most $\hat{\sigma}^2$
1: Set $\mathbf{g}_0 \leftarrow$ `BiasedStochasticGradient`$(\mathbf{x}, \hat{\sigma}/10)$
2: Randomly sample $j \sim \text{Geom}\left(\frac{1}{2}\right) \in \mathbb{N}$
3: $\mathbf{g}_j \leftarrow$ `BiasedStochasticGradient`$(\mathbf{x}, 2^{-3j/4}\hat{\sigma}/10)$
4: $\mathbf{g}_{j-1} \leftarrow$ `BiasedStochasticGradient`$(\mathbf{x}, 2^{-3(j-1)/4}\hat{\sigma}/10)$
5: $\mathbf{g} \leftarrow \mathbf{g}_0 + 2^j(\mathbf{g}_j - \mathbf{g}_{j-1})$
5: Return $\mathbf{g}$

---

**Lemma A.1.** *Given access to an algorithm* `BiasedStochasticGradient` *that outputs a random vector* $\mathbf{v}$ *such that* $\mathbb{E}\|\mathbf{v} - \nabla f(\mathbf{x})\| \leq \hat{\sigma}^2$ *with a cost* $\tilde{\mathcal{O}}\left(\frac{C}{\hat{\sigma}}\right)$, *the algorithm* `UnbiasedStochasticGradient` *outputs a vector* $\mathbf{g}$ *such that* $\mathbb{E}[\mathbf{g}] = \nabla f(\mathbf{x})$ *and* $\mathbb{E}\|\mathbf{g} - \nabla f(\mathbf{x})\| \leq \hat{\sigma}^2$ *with an expected cost* $\tilde{\mathcal{O}}\left(\frac{C}{\hat{\sigma}}\right)$.

*Proof.* We repeat the proof in Sidford & Zhang (2023).

$$\mathbf{g} = \mathbf{g}_0 + 2^J(\mathbf{g}_J - \mathbf{g}_{J-1}), \qquad J \sim \text{Geom}\left(\frac{1}{2}\right) \in \mathbb{N}. \tag{21}$$

Given that $\Pr(J = j) = 2^{-j}$, we have

$$\mathbb{E}[\mathbf{g}] = \mathbb{E}[\mathbf{g}_0] + \sum_{j=1}^{\infty} \Pr(J = j)2^j(\mathbb{E}[\mathbf{g}_j] - \mathbb{E}[\mathbf{g}_{j-1}]) = \mathbb{E}[\mathbf{g}_\infty] = \nabla f(\mathbf{x}). \tag{22}$$

As for the variance, using the inequality $(a + b)^2 \leq 2a^2 + 2b^2$, we have

$$\mathbb{E}\|\mathbf{g} - \nabla f(\mathbf{x})\|^2 \leq 2\mathbb{E}\|\mathbf{g} - \mathbf{g}_0\|^2 + 2\mathbb{E}\|\mathbf{g}_0 - \nabla f(\mathbf{x})\|^2 \tag{23}$$

where

$$\mathbb{E}\|\mathbf{g} - \mathbf{g}_0\|^2 = \sum_{j=1}^{\infty} \Pr(J = j)2^{2j}\mathbb{E}\|\mathbf{g}_j - \mathbf{g}_{j-1}\|^2 = \sum_{j=1}^{\infty} 2^j\mathbb{E}\|\mathbf{g}_j - \mathbf{g}_{j-1}\|^2, \tag{24}$$

and for each $j$ we have

$$\mathbb{E}\|\mathbf{g}_j - \mathbf{g}_{j-1}\|^2 \leq 2\mathbb{E}\|\mathbf{g}_j - \nabla f(\mathbf{x})\|^2 + 2\mathbb{E}\|\mathbf{g}_{j-1} - \nabla f(\mathbf{x})\|^2. \tag{25}$$

By assumption on `BiasedStochasticGradient`,

$$\mathbb{E}\|\mathbf{g}_j - \nabla f(\mathbf{x})\|^2 \leq \frac{\hat{\sigma}^2}{100 \cdot 2^{3j/2}}, \quad \forall j \geq 0, \tag{26}$$

which leads to

$$\mathbb{E}\|\mathbf{g}_j - \mathbf{g}_{j-1}\|^2 \leq \frac{\hat{\sigma}^2}{50 \cdot 2^{3(j-1)/2}} + \frac{\hat{\sigma}^2}{50 \cdot 2^{3j/2}} \leq \frac{\hat{\sigma}^2}{10 \cdot 2^{3j/2}}, \tag{27}$$

and

$$\mathbb{E}\|\mathbf{g} - \mathbf{g}_0\|^2 = \frac{\hat{\sigma}^2}{10}\sum_{j=1}^{\infty} \frac{1}{2^{j/2}} \leq \frac{1}{3}\hat{\sigma}^2. \tag{28}$$

Hence,

$$\mathbb{E}\|\mathbf{g} - \nabla f(\mathbf{x})\|^2 \leq 2\mathbb{E}\|\mathbf{g} - \mathbf{g}_0\|^2 + 2\mathbb{E}\|\mathbf{g}_0 - \nabla f(\mathbf{x})\|^2 \leq \hat{\sigma}^2, \tag{29}$$

Moreover, the expected cost is

$$\tilde{\mathcal{O}}\left(\frac{C}{\hat{\sigma}}\right) \cdot \left(1 + \sum_{j=1}^{\infty} \Pr\{J = j\} \cdot \left(2^{3j/4} + 2^{3(j-1)/4}\right)\right) = \tilde{\mathcal{O}}\left(\frac{C}{\hat{\sigma}}\right). \tag{30}$$

$\square$

### A.3 QUANTUM MEAN ESTIMATION

We assume access to the following oracle for a probability distribution.

**Definition A.2** (Quantum Sampling Oracle). Quantum sampling oracle $O_X$ of a random variable $X \in \Omega$ is given by $O_X |0\rangle |0\rangle \mapsto \sum_{X \in \Omega} \sqrt{\Pr(X)} |X\rangle |\text{garbage}(X)\rangle$.

Here, the second register contains $|\text{garbage}(X)\rangle$, which depends on $X$. The state in the (auxiliary) garbage register is usually generated in some intermediate step of computing $X$ in the first register. It is important to note that the state in this quantum sampling oracle differs from the coherent quantum sample state, as the former is entangled and we cannot simply discard the garbage register.

Quantum mean estimation is a technique to estimate the mean of a $d$-dimensional random variable $X$ up to $\epsilon$ accuracy using $\tilde{\mathcal{O}}(d^{1/2}/\epsilon)$ queries, which is a quadratic improvement in $\epsilon$ compared to classical algorithms Cornelissen et al. (2022). Although the quantum mean estimation algorithm is biased, Sidford & Zhang (2023) developed an unbiased quantum mean estimation algorithm. Specifically, for a multi-dimensional variable with mean $\mu$ and variance $\sigma^2$, unbiased quantum mean estimation outputs an estimate $\hat{\mu}$ such that $\mathbb{E}[\hat{\mu}] = \mu$ and $\mathbb{E}[\|\hat{\mu} - \mu\|^2] \leq \hat{\sigma}^2$ using $\tilde{\mathcal{O}}(d^{1/2}\sigma/\hat{\sigma})$ queries.

**Lemma A.3** (Unbiased Quantum Mean Estimation Sidford & Zhang (2023)). *For a $d$-dimensional random variable $X$ with $\text{Var}[X] \leq \sigma^2$ and some $\hat{\sigma} \geq 0$, suppose we are given access to its quantum sampling oracle $O_X$ (as in Definition A.2). Then, there is a procedure* QuantumMeanEstimation$(O_X, \hat{\sigma})$ *that uses* $\tilde{\mathcal{O}}\left(\frac{d^{1/2}\sigma}{\hat{\sigma}}\right)$ *queries to $O_X$ and outputs an unbiased estimate $\hat{\mu}$ of the expectation $\mu$ satisfying* $\text{Var}[\hat{\mu}] \leq \hat{\sigma}^2$.

### A.4 OVERVIEW OF JORDAN'S ALGORITHM

Jordan's algorithm Jordan (2005) approximates the gradient using a finite difference formula on a small grid around the point of interest and encodes the estimate into the quantum phase. Then, the algorithm applies an inverse quantum Fourier transform to estimate the gradient. Although Jordan's original analysis implicitly assumes that higher-order derivatives of the function are negligible, Gilyén, Arunachalam, and Wiebe Gilyén et al. (2019) analyzed the algorithm and extended it to handle functions in the Gevrey class, using central difference formulas and a binary oracle model commonly encountered in variational quantum algorithms. The closest analysis of Jordan's algorithm to our setting was provided by Chakrabarti et al. (2020), who demonstrated that Algorithm 4 achieves constant query complexity for functions with Lipschitz gradients, provided that the function values can be queried with high precision.

The following lemma from Chakrabarti et al. (2020) demonstrates that Algorithm 4 achieves $\tilde{O}(1)$ query complexity for evaluating the gradient of a $\beta$-smooth function with high probability.

**Lemma A.4** (Lemma 2.2 in Chakrabarti et al. (2020)). *Let $f\colon \mathbb{R}^d \to \mathbb{R}$ be a function that is accessible via an evaluation oracle with error at most $\epsilon$. Assume that $\|\nabla f\| \leq L$ and $f$ is $\beta$-smooth in $B_\infty(x, 2\sqrt{\epsilon/\beta})$. Let $\tilde{\mathbf{g}}$ be the output of* QuantumGradient$(f, \epsilon, M, \beta, \mathbf{x}_0)$ *(as defined in Algorithm 4). Then:*

$$\Pr\left[|\tilde{\mathbf{g}}_i - \nabla f(\mathbf{x})_i| > 1500\sqrt{d\epsilon\beta}\right] < \frac{1}{3}, \quad \forall i \in [d]. \tag{31}$$

Although Algorithm 4 results in an accurate estimate for the gradient with high probability, it is possible to run the algorithm multiple times and take the coordinate-wise median of the outputs to obtain a smooth estimate for the gradient (Lemma 2.3 in Chakrabarti et al. (2020)) when the norm of the gradient is bounded. To estimate the gradient up to $\delta$ error (in $L2$ norm), it is required to have an evaluation oracle with error at most $\mathcal{O}(\delta^2/d^2)$ which might not be feasible if the noisy evaluation oracle is stochastic.

Our algorithms work in the stochastic setting where we prove that we can create an accurate evaluation oracle under Assumptions 2.1 and 2.2. Furthermore, the function $f$ needs to be smooth; however, under Assumptions 2.1 and 2.5 the smoothness constant is not bounded and this might cause unbounded error. We propose a robust version of Algorithm 4 so that we can still estimate the gradient accurately (See the step-by-step description in Section 2.2). We also note that the oracle $O_F$ is known as the phase oracle. Our oracle (Eq. (6)) can be converted to phase oracle efficiently.

---

**Algorithm 4** QuantumGradient($f, \epsilon, L, \beta, x_0$)

---

0: **Input**: Function $f$, evaluation error $\epsilon$, gradient norm bound $L$, smoothness parameter $\beta$, and point $\mathbf{x}_0$.
   Define
   - $l = 2\sqrt{\epsilon/\beta d}$ to be the size of the grid used,
   - $b \in \mathbb{N}$ such that $\frac{24\pi\sqrt{d\epsilon\beta}}{L} \leq \frac{1}{2^b} = \frac{1}{N} \leq \frac{48\pi\sqrt{d\epsilon\beta}}{L}$,
   - $b_0 \in \mathbb{N}$ such that $\frac{N\epsilon}{2Ll} \leq \frac{1}{2^{b_0}} = \frac{1}{N_0} \leq \frac{N\epsilon}{Ll}$,
   - $F(x) = \frac{N}{2Ll}[f(x_0 + \frac{l}{N}(x - N/2)) - f(x_0)]$, and,
   - $\gamma : \{0, 1, \ldots, N-1\} \to G := \{-N/2, -N/2+1, \ldots, N/2-1\}$ s.t. $\gamma(x) = x - N/2$.

   Let $O_F$ denote a unitary operation acting as $O_F \ket{x} = e^{2\pi i \tilde{F}(x)} \ket{x}$, where $|\tilde{F}(x) - F(x)| \leq \frac{1}{N_0}$, with $x$ represented using $b$ bits and $\tilde{F}(x)$ represented using $b_0$ bits.

1: Start with $n$ $b$-bit registers set to 0 and Hadamard transform each to obtain

$$\frac{1}{\sqrt{N^n}} \sum_{x_1,\ldots,x_n \in \{0,1,\ldots,N-1\}} \ket{x_1, \ldots, x_n} ;$$

2: Perform the operation $O_F$ and the map $\ket{x} \mapsto \ket{\gamma(x)}$ to obtain

$$\frac{1}{N^{n/2}} \sum_{\mathbf{g} \in G^n} e^{2\pi i \tilde{F}(\mathbf{g})} \ket{\mathbf{g}} ;$$

3: Apply the inverse QFT over $G$ to each of the registers
4: Measure the final state to get $k_1, k_2, \ldots, k_n$ and report $\tilde{\mathbf{g}} = \frac{2L}{N}(k_1, k_2, \ldots, k_n)$ as the result.

---

# B   PROOFS FOR HAMILTONIAN MONTE CARLO IN STRONGLY CONVEX CASE

We start with the following result in Zou & Gu (2021) that quantifies the convergence of the stochastic gradient Hamiltonian Monte Carlo algorithm in Wasserstein distance.

**Theorem B.1** (Theorem 4.4 in Zou & Gu (2021)). *Under Assumptions 2.2 and 3.1, let $D = \|\mathbf{x}^0 - \arg\min_{\mathbf{x}}(f(\mathbf{x}))\|$ and $\mu_T$ be the distribution of the iterate $\mathbf{x}^T$, then if the step size satisfies $\eta = O(L^{1/2}\sigma^{-2}\kappa^{-1} \wedge L^{-1/2})$ and $K = 1/(4\sqrt{L}\eta)$, the output of HMC satisfies*

$$\mathbf{W}_2(\mu_T, \pi) \leq (1 - (128\kappa)^{-1})^{\frac{T}{2}}(2D + 2d/\mu)^{1/2} + \Gamma_1\eta^{1/2} + \Gamma_2\eta, \tag{32}$$

*where $\Gamma_1^2 = O(L^{-3/2}\sigma^2\kappa^2)$ and $\Gamma_2^2 = O(\kappa^2(LD + \kappa d + L^{-1/2}\sigma^2\eta))$ where $\sigma^2 = \max_{t \leq T} \mathbb{E}\|\mathbf{g}(\mathbf{x}_k, \boldsymbol{\xi}_k) - \nabla f(\mathbf{x}_k)\|^2$ is the upper bound on the variance of the gradients in the trajectory of* SG-HMC *algorithm.*

This is a generic result that applies to any HMC algorithm under Assumptions 2.2 and 3.1 that uses stochastic gradients with variance upper bounded by $\sigma^2$. Note that we do not assume a uniform upper bound for $\sigma$ that is independent of problem parameters. Instead, the variance upper bound depends on the trajectory of the algorithm, which can be characterized using theoretical analysis. The analysis in Zou & Gu (2021) assumes Lipschitzness of the stochastic gradients, and this follows from the smoothness of $f$ in this case as we set the final error in our gradient estimation to $\epsilon$ so that the gradients are very close to gradient of $\nabla f$.

## B.1   PROOF OF QZ-HMC

**Theorem 3.2** (Main Theorem for QZ-HMC). *Let $\mu_k$ be the distribution of $\mathbf{x}_k$ in* QZ-HMC *algorithm. Suppose that $f$ satisfies Assumption 3.1. Given that the initial point $\mathbf{x}_0$ satisfies $\|\mathbf{x}_0 - \arg\min_{\mathbf{x}} f(\mathbf{x})\| \leq \frac{d}{\mu}$, if we set the step size $\eta = \mathcal{O}\left(\frac{\epsilon}{d^{1/2}\kappa^{3/2}}\right)$, $S = \tilde{\mathcal{O}}\left(\frac{Ld^{1/2}\kappa^{3/2}}{\epsilon}\right)$, $T = \tilde{\mathcal{O}}(1)$, and $\hat{\sigma}^2 = \mathcal{O}\left(\frac{L^{3/2}d^{1/2}\epsilon}{\kappa^{3/2}}\right)$, we have*

$$\mathbf{W}_2(\mu_{ST}, \pi) \leq \epsilon.$$

In addition, under Assumptions 2.1 and 2.2, the query complexity to the stochastic evaluation oracle is $\tilde{\mathcal{O}}\left(\frac{d^{5/4}\sigma}{\epsilon^{3/2}}\right)$ or under Assumptions 2.1 and 2.5 the query complexity to the stochastic evaluation oracle is $\tilde{\mathcal{O}}\left(\frac{d^{3/4}\sigma}{\epsilon^{3/2}}\right)$.

*Proof.* By Theorem B.1 for $\eta = \mathcal{O}(L^{1/2}\sigma^{-2}\kappa^{-1} \wedge L^{-1/2})$ and $K = \frac{1}{4\sqrt{L}\eta}$, we have

$$\mathrm{W}_2(\mu_T, \pi) \leq (1 - (128\kappa)^{-1})^{\frac{T}{2}}(2D + 2d/\mu)^{1/2} + \Gamma_1\eta^{1/2} + \Gamma_2\eta, \tag{33}$$

where

$$\Gamma_1 = \mathcal{O}\left(L^{-3/2}\hat{\sigma}^2\kappa^2\right), \tag{34}$$

$$\Gamma_2 = \mathcal{O}\left(\kappa^3 d\right). \tag{35}$$

The first term in error is $\mathcal{O}(\epsilon)$ when $T = \tilde{\mathcal{O}}(\log(1/\epsilon))$. The last two terms become $\mathcal{O}\left(L^{-3/4}\hat{\sigma}\eta^{1/2} + d^{1/2}\kappa^{3/2}\eta\right)$. For $\hat{\sigma} = \mathcal{O}(L^{3/4}d^{1/4}\kappa^{-3/4}\epsilon^{1/2} \wedge \sigma)$ and $\eta = \mathcal{O}(\epsilon d^{-1/2}\kappa^{-3/2})$, the bias term becomes $\mathcal{O}(\epsilon)$. Then, under Assumptions 2.1 and 2.2, the number of calls to evaluation oracle scale as $\tilde{\mathcal{O}}(d^{1/2}\kappa^{3/2}\epsilon^{-1} + \sigma d^{3/4}\kappa^{3/4}\epsilon^{-3/2}) = \tilde{\mathcal{O}}(d^{3/4}\kappa^{3/4}\epsilon^{-3/2})$. Similarly, under Assumptions 2.1 and 2.5 the evaluation complexity is $\tilde{\mathcal{O}}(\sigma d^{5/4}\kappa^{3/4}\epsilon^{-3/2})$. $\square$

## C  PROOFS FOR LSI CASE

**Lemma C.1** (Stochastic-LMC One Step Convergence). *Let $\mu_k$ be the distribution of the iterate $\mathbf{x}_k$, then if the step size satisfies $\eta = \frac{2}{3\alpha}$,*

$$\mathrm{KL}(\mu_{k+1}||\pi) \leq e^{-3\alpha\eta/2}\left[\left(1 + \frac{32\eta^3 L^4}{\alpha}\right)\mathrm{KL}(\mu_k||\pi) + 6\eta\sigma_k^2 + 16\eta^2 dL^2\right], \tag{36}$$

*where $\sigma_k^2 = \mathbb{E}_{\mathbf{x}_k, \boldsymbol{\xi}_k}\|\mathbf{g}(\mathbf{x}_k, \boldsymbol{\xi}_k) - \nabla f(\mathbf{x}_k)\|^2$.*

*Proof.* We compare one step of LMC starting at $\mathbf{x}_k$ with stochastic gradients $\mathbf{g}(\mathbf{x}_k, \boldsymbol{\xi}_k)$ to the output of continuous Langevin SDE (Eq. (2)) starting at $\mathbf{x}_k$ with true gradient $\nabla f(\mathbf{x}_t)$ after time $\eta$. This technique has been used to establish the convergence of unadjusted Langevin algorithm with full gradients under isoperimetry by Vempala & Wibisono (2019). We extend the analysis by Vempala & Wibisono (2019) to the stochastic gradient LMC. Assume that the initial point $\mathbf{x}_k$ and $\mathbf{g}(\mathbf{x}_k, \boldsymbol{\xi}_k)$ obey the joint distribution $\mu_0$. The randomness on $\mathbf{g}(\mathbf{x}_k, \boldsymbol{\xi}_k)$ depends both on the randomness on $\mathbf{x}_k$ and the randomness in the quantum mean estimation algorithm. Then, one step update of LMC algorithm with stochastic gradient yields,

$$\mathbf{x}_{k+1} = \mathbf{x}_k - \eta\mathbf{g}(\mathbf{x}_k, \boldsymbol{\xi}_k) + \sqrt{2\eta}\boldsymbol{\epsilon}_k.$$

Alternatively, $\mathbf{x}_{k+1}$ can be written as the solution of the following SDE at time $t = \eta$,

$$d\mathbf{x}_t = -\mathbf{g}_k dt + \sqrt{2}d\boldsymbol{W}_t$$

where $\mathbf{g}_k = \mathbf{g}(\mathbf{x}_k, \boldsymbol{\xi}_k)$ and $\boldsymbol{W}_t$ is the standard Brownian motion starting at $\boldsymbol{W}_0 = 0$. Let $\mu_t(\mathbf{x}_k, \mathbf{g}_k, \mathbf{x}_t)$ be the joint distribution of $\mathbf{x}_k, \mathbf{g}_k$, and $\mathbf{x}_t$ at time $t$. Each expectation in the proof is over this joint distribution unless specified otherwise.

Consider the following stochastic differential equation

$$d\boldsymbol{X} = \boldsymbol{v}(\boldsymbol{X})dt + \sqrt{2}d\boldsymbol{W},$$

where $\boldsymbol{v}$ is a smooth vector field and $\boldsymbol{W}$ is the Brownian motion with $\boldsymbol{W}_0 = 0$. The Fokker-Planck equation describes the evolution of probability density function $\mu_t$ as follows:

$$\frac{\partial\mu_t}{\partial t} = -\nabla \cdot (\mu_t\boldsymbol{v}) + \Delta\mu_t, \tag{37}$$

where $\nabla\cdot$ is the divergence operator and $\Delta$ is the Laplacian. Then, the Fokker Planck equation gives the following evolution for the marginal density $\mu_t(\mathbf{x}|\mathbf{x}_k, \mathbf{g}_k) = \mu_t(\mathbf{x}_t = \mathbf{x}|\mathbf{x}_k, \mathbf{g}_k)$,

$$\frac{\partial \mu_t(\mathbf{x}|\mathbf{x}_k, \mathbf{g}_k)}{\partial t} = \nabla \cdot (\mu_t(\mathbf{x}|\mathbf{x}_k, \mathbf{g}_k)\mathbf{g}_k) + \Delta \mu_t(\mathbf{x}|\mathbf{x}_k, \mathbf{g}_k). \tag{38}$$

Taking the expectation over both sides with respect to $(\mathbf{x}_k, \mathbf{g}_k) \sim \mu_0$,

$$\frac{\partial \mu_t(\mathbf{x})}{\partial t} = \mathbb{E}_{(\mathbf{x}_k, \mathbf{g}_k) \sim \mu_0}[\nabla \cdot (\mu_t(\mathbf{x}|\mathbf{x}_k)\mathbf{g}_k)] + \mathbb{E}_{(\mathbf{x}_k, \mathbf{g}_k) \sim \mu_0}[\Delta \mu_t(\mathbf{x}|\mathbf{x}_k)] \tag{39}$$

$$= \int_{\mathbb{R}^d} \nabla \cdot (\mu_t(\mathbf{x}|\mathbf{x}_k, \mathbf{g}_k)\mathbf{g}_k)\mu_0(\mathbf{x}_k, \mathbf{g}_k)d\mathbf{x}_k d\mathbf{g}_k + \int_{\mathbb{R}^d} \Delta \mu_t(\mathbf{x}|\mathbf{x}_k, \mathbf{g}_k)\mu_0(\mathbf{x}_k, \mathbf{g}_k)d\mathbf{x}_k d\mathbf{g}_k \tag{40}$$

$$= \int_{\mathbb{R}^d} \nabla \cdot (\mu_t(\mathbf{x})\mu(\mathbf{x}_k, \mathbf{g}_k|\mathbf{x}_t = \mathbf{x})\mathbf{g}_k)d\mathbf{x}_k d\mathbf{g}_k + \Delta \mu_t(\mathbf{x}) \tag{41}$$

$$= \nabla \cdot \left( \mu_t(\mathbf{x})\mathbb{E}[\mathbf{g}_k - \nabla f(\mathbf{x}_k)|\mathbf{x}_t = \mathbf{x}] + \mu_t(\mathbf{x})\nabla \log\left(\frac{\mu_t(\mathbf{x})}{\pi(\mathbf{x})}\right) \right). \tag{42}$$

Consider the time derivative of KL divergence between $\mu_t$ and $\pi$,

$$\frac{d}{dt}\mathrm{KL}(\mu_t||\pi) = \frac{d}{dt}\int_{\mathbb{R}^d} \mu_t(\mathbf{x})\log\left(\frac{\mu_t(\mathbf{x})}{\pi(\mathbf{x})}\right)d\mathbf{x} \tag{43}$$

$$= \int_{\mathbb{R}^d} \frac{\partial \mu_t(\mathbf{x})}{\partial t}\log\left(\frac{\mu_t(\mathbf{x})}{\pi(\mathbf{x})}\right)d\mathbf{x}_t + \int_{\mathbb{R}^d} \mu_t(\mathbf{x})\frac{\partial}{\partial t}\log\left(\frac{\mu_t(\mathbf{x})}{\pi(\mathbf{x})}\right)d\mathbf{x} \tag{44}$$

$$= \int_{\mathbb{R}^d} \frac{\partial \mu_t(\mathbf{x})}{\partial t}\log\left(\frac{\mu_t(\mathbf{x})}{\pi(\mathbf{x})}\right)d\mathbf{x}_t + \int_{\mathbb{R}^d} \frac{\partial \mu_t(\mathbf{x})}{\partial t}d\mathbf{x} \tag{45}$$

$$= \int_{\mathbb{R}^d} \frac{\partial \mu_t(\mathbf{x})}{\partial t}\log\left(\frac{\mu_t(\mathbf{x})}{\pi(\mathbf{x})}\right)d\mathbf{x}_t. \tag{46}$$

The last term in the third equality vanishes since the $\mu_t$ is probability distribution and its $L_1$ norm is always 1. Then the KL divergence evolves as

$$\frac{d}{dt}\mathrm{KL}(\mu_t||\pi) = \int_{\mathbb{R}^d} \nabla \cdot \left( \mu_t(\mathbf{x})\mathbb{E}[\mathbf{g}_k - \nabla f(\mathbf{x})|\mathbf{x}_t = \mathbf{x}] + \mu_t(\mathbf{x})\nabla \log\left(\frac{\mu_t(\mathbf{x})}{\pi(\mathbf{x})}\right) \right)\log\left(\frac{\mu_t(x)}{\pi(\mathbf{x})}\right)d\mathbf{x} \tag{47}$$

$$= -\int_{\mathbb{R}^d} \mu_t(\mathbf{x})\left\langle \mathbb{E}[\mathbf{g}_k - \nabla f(\mathbf{x})|\mathbf{x}_t = \mathbf{x}] + \nabla \log\left(\frac{\mu_t(\mathbf{x})}{\pi(\mathbf{x})}\right), \nabla \log\left(\frac{\mu_t(\mathbf{x})}{\pi(\mathbf{x})}\right) \right\rangle d\mathbf{x} \tag{48}$$

$$= -\int_{\mathbb{R}^d} \mu_t(\mathbf{x})\left\| \nabla \log\left(\frac{\mu_t(\mathbf{x})}{\pi(\mathbf{x})}\right) \right\|^2 d\mathbf{x} + \mathbb{E}\left\langle \nabla f(\mathbf{x}_t) - \mathbf{g}_k, \nabla \log\left(\frac{\mu_t(\mathbf{x})}{\pi(\mathbf{x})}\right) \right\rangle. \tag{49}$$

The second term can be bounded as follows:

$$\mathbb{E}\left\langle \nabla f(\mathbf{x}_t) - \mathbf{g}_k, \nabla \log\left(\frac{\mu_t(\mathbf{x})}{\pi(\mathbf{x})}\right)\right\rangle \leq \mathbb{E}\left[\|\nabla f(\mathbf{x}_t) - \mathbf{g}_k\|^2 + \frac{1}{4}\left\|\nabla \log\left(\frac{\mu_t(\mathbf{x})}{\pi(\mathbf{x})}\right)\right\|^2\right] \tag{50}$$

$$= \mathbb{E}\|\nabla f(\mathbf{x}_t) - \mathbf{g}_k\|^2 + \frac{1}{4}\mathrm{FI}(\mu_t||\pi) \tag{51}$$

$$= \mathbb{E}\|\nabla f(\mathbf{x}_t) - \nabla f(\mathbf{x}_k) + \nabla f(\mathbf{x}_k) - \mathbf{g}_k\|^2 + \frac{1}{4}\mathrm{FI}(\mu_t||\pi) \tag{52}$$

$$\leq 2\mathbb{E}\|\nabla f(\mathbf{x}_t) - \nabla f(\mathbf{x}_k)\|^2 + 2\mathbb{E}_{\mu_t(\mathbf{x}_t, \mathbf{x}_k)}\|\nabla f(\mathbf{x}_k) - \mathbf{g}_k\|^2 \tag{53}$$

$$+ \frac{1}{4}\mathrm{FI}(\mu_t||\pi). \tag{54}$$

The first inequality holds since $\langle a, b\rangle \leq a^2 + \frac{b^2}{4}$. The last line follows from Young's inequality. Furthermore, using Lipschitzness of gradients of $f$, we have

$$\mathbb{E}\|\nabla f(\mathbf{x}_t) - \nabla f(\mathbf{x}_k)\|^2 \leq L^2 \mathbb{E}\|\mathbf{x}_t - \mathbf{x}_k\|^2 \tag{55}$$

$$\leq L^2 \mathbb{E}\| - t\mathbf{g}_k + \sqrt{2t}\boldsymbol{\epsilon}_k\|^2 \tag{56}$$

$$= t^2 L^2 \mathbb{E}_{\mu_0}\|\mathbf{g}_k\|^2 + 2tdL^2. \tag{57}$$

Plugging back these into the time derivative of KL divergence, we have

$$\frac{d}{dt}\mathrm{KL}(\mu_t||\pi) \leq -\frac{3}{4}\mathrm{FI}(\mu_t||\pi) + 2t^2 L^2 \mathbb{E}_{\mu_0}\|\mathbf{g}_k\|^2 + 2\mathbb{E}_{\mu_0}\|\nabla f(\mathbf{x}_k) - \mathbf{g}_k\|^2 + 4tdL^2 \tag{58}$$

$$\leq -\frac{3}{4}\mathrm{FI}(\mu_t||\pi) + (4t^2 L^2 + 2)\mathbb{E}_{\mu_0}\|\nabla f(\mathbf{x}_k) - \mathbf{g}_k\|^2 + 4t^2 L^2 \mathbb{E}_{\mu_0}\|\nabla f(\mathbf{x}_k)\|^2 + 4tdL^2. \tag{59}$$

The third term can be bounded as follows: We choose an optimal coupling $\mathbf{x}_k \sim \mu_0(\mathbf{x}_k)$ and $\mathbf{x}^\star \sim \pi$ so that $\mathbb{E}\|\mathbf{x}_k - \mathbf{x}^\star\| = \mathrm{W}_2(\mu_0, \pi)^2$, then using Young's inequality and smoothness of $f$,

$$\mathbb{E}_{\mu_0}\|\nabla f(\mathbf{x}_k)\|^2 \leq 2\mathbb{E}_{\mu_0}\|\nabla f(\mathbf{x}_k) - \nabla f(\mathbf{x}^\star)\|^2 + 2\mathbb{E}_{\mu_0}\|\nabla f(\mathbf{x}^\star)\|^2 \tag{60}$$

$$\leq 2L^2 \mathbb{E}_{\mu_0}\|\mathbf{x}_k - \mathbf{x}_0\|^2 + 2\mathbb{E}_{\mu_0}\|\nabla f(\mathbf{x}^\star)\|^2 \tag{61}$$

$$\leq 2L^2 \mathrm{W}_2(\mu_0, \pi)^2 + 2dL \tag{62}$$

$$\leq \frac{4L^2}{\alpha}\mathrm{KL}(\mu_0||\pi) + 2dL. \tag{63}$$

The last inequality follows from Talgrand's inequality. Hence for $t \leq \eta$ and $\eta \leq \frac{1}{2L}$, we have

$$\frac{d}{dt}\mathrm{KL}(\mu_t||\pi) \leq -\frac{3}{4}\mathrm{FI}(\mu_t||\pi) + (4t^2 L^2 + 2)\mathbb{E}_{\mu_0}\|\nabla f(\mathbf{x}_k) - \mathbf{g}_k\|^2 + \frac{16t^2 L^4}{\alpha}\mathrm{KL}(\mu_0||\pi) + 4tdL^2 + 8t^2 dL^3 \tag{64}$$

$$\leq -\frac{3\alpha}{2}\mathrm{KL}(\mu_t||\pi) + (4t^2 L^2 + 2)\mathbb{E}_{\mu_0}\|\nabla f(\mathbf{x}_k) - \mathbf{g}_k\|^2 + \frac{16t^2 L^4}{\alpha}\mathrm{KL}(\mu_0||\pi) + 4tdL^2 + 8t^2 dL^3 \tag{65}$$

$$\leq -\frac{3\alpha}{2}\mathrm{KL}(\mu_t||\pi) + 3\mathbb{E}_{\mu_0}\|\nabla f(\mathbf{x}_k) - \mathbf{g}_k\|^2 + \frac{16\eta^2 L^4}{\alpha}\mathrm{KL}(\mu_0||\pi) + 8\eta dL^2 \tag{66}$$

$$\leq -\frac{3\alpha}{2}\mathrm{KL}(\mu_t||\pi) + 3\sigma_k^2 + \frac{16\eta^2 L^4}{\alpha}\mathrm{KL}(\mu_0||\pi) + 8\eta dL^2. \tag{67}$$

The second inequality is due to Eq. (16). Equivalently, we can write,

$$\frac{d}{dt}(e^{3\alpha t/2}\mathrm{KL}(\mu_t||\pi)) \leq e^{3\alpha t/2}\left(3\sigma_k^2 + \frac{16\eta^2 L^4}{\alpha}\mathrm{KL}(\mu_0||\pi) + 8\eta dL^2\right). \tag{68}$$

Integrating from $t = 0$ to $t = \eta$ gives,

$$e^{3\alpha\eta/2}\mathrm{KL}(\mu_\eta||\pi) - \mathrm{KL}(\mu_0||\pi) \leq 6\eta\sigma_k^2 + \frac{32\eta^3 L^4}{\alpha}\mathrm{KL}(\mu_0||\pi) + 16\eta^2 dL^2 \tag{69}$$

for $\eta \leq \frac{2}{3\alpha}$. Rearranging the terms,

$$\mathrm{KL}(\mu_\eta || \pi) \leq e^{-3\alpha\eta/2} \left[ \left( 1 + \frac{32\eta^3 L^4}{\alpha} \right) \mathrm{KL}(\mu_0 || \pi) + 6\eta\sigma_k^2 + 16\eta^2 dL^2 \right]. \tag{70}$$

Renaming $\mu_0 = \mu_k$ and $\mu_\eta = \mu_{k+1}$, we obtain the result in the statement.

$\square$

The statement in Lemma C.1 is generic and can be applied to any LMC algorithm with stochastic gradients with bounded variance on the trajectory of the algorithm. Note that this is different from assuming that the variance is uniformly upper bounded. Instead, we set inner loop and variance reduction parameters so that the variance does not explode along the trajectory of the algorithm.

## C.1 PROOF OF QZ-LMC

**Theorem 3.4** (Main Theorem for `QZ-LMC`). *Under Assumption 3.3, let $\mu_k$ be the distribution of $\mathbf{x}_k$ in* `QZ-LMC` *algorithm. Then, if we set the step size $\eta = \mathcal{O}\left(\frac{\epsilon\alpha}{dL^2}\right)$, $K = \tilde{\mathcal{O}}\left(\frac{dL^2 \log(\mathrm{KL}(\mu_0||\pi))}{\epsilon\alpha^2}\right)$, and $\hat{\sigma}^2 = \mathcal{O}(\alpha\epsilon)$, we have*

$$\left\{ \mathrm{KL}(\mu_K || \pi), \mathrm{TV}(\mu_K, \pi)^2, \frac{\alpha}{2} \mathrm{W}_2(\mu_K, \pi)^2 \right\} \leq \epsilon.$$

*In addition, under Assumptions 2.1 and 2.2, the query complexity to the stochastic evaluation oracle is $\tilde{\mathcal{O}}\left(\frac{d^2 L^2 \sigma}{\alpha^{5/2}\epsilon^{3/2}}\right)$, or under Assumptions 2.1 and 2.5 the query complexity to the stochastic evaluation oracle is $\tilde{\mathcal{O}}\left(\frac{d^{3/2} L^2 \sigma}{\alpha^{5/2}\epsilon^{3/2}}\right)$.*

*Proof.* By Lemma C.1, one-step equation can be written as

$$\mathrm{KL}(\mu_{k+1} || \pi) \leq e^{-3\alpha\eta/2} \left[ \left( 1 + \frac{32\eta^3 L^4}{\alpha} \right) \mathrm{KL}(\mu_k || \pi) + 6\eta\hat{\sigma}^2 + 16\eta^2 dL^2 \right] \tag{71}$$

$$\leq e^{-\alpha\eta} \mathrm{KL}(\mu_k || \pi) + 6\eta\hat{\sigma}^2 + 16\eta^2 dL^2. \tag{72}$$

Since for $\eta \leq \frac{\alpha}{8L^2}$, $1 + \frac{32\eta^3 L^4}{\alpha} \leq 1 + \frac{\alpha\eta}{2} \leq e^{\alpha\eta/2}$. Unrolling the recursion, we have

$$\mathrm{KL}(\mu_k || \pi) \leq e^{-\alpha\eta k} \mathrm{KL}(\mu_0 || \pi) + \frac{6\eta\hat{\sigma}^2 + 16\eta^2 dL^2}{1 - e^{-\alpha\eta}} \tag{73}$$

$$\leq e^{-\alpha\eta k} \mathrm{KL}(\mu_0 || \pi) + \frac{8\hat{\sigma}^2 + 32\eta dL^2}{\alpha} \tag{74}$$

$$\leq e^{-\alpha\eta k} \mathrm{KL}(\mu_0 || \pi) + \frac{8\hat{\sigma}^2 + 32\eta dL^2}{\alpha}. \tag{75}$$

The second inequality is due to the fact that for $\eta \leq \frac{\alpha}{8L^2}$, $1 - e^{-\alpha\eta} \geq \frac{3}{4}\alpha\eta$ when $\alpha\eta \leq \frac{1}{4}$. We set $\eta \leq \frac{\epsilon\alpha}{128dL^2}$ and $\hat{\sigma}^2 \leq \frac{\alpha\epsilon}{32}$ and $k \geq \frac{1}{\alpha\eta} \log \left( \frac{2\mathrm{KL}(\mu_0||\pi)}{\epsilon} \right)$ so that $\mathrm{KL}(\mu_k || \pi) \leq \epsilon$. The number of calls to the stochastic evaluation oracle under Assumptions 2.1 and 2.2 to achieve $\hat{\sigma}^2 \leq \frac{\alpha\epsilon}{32}$ at each iteration is $\mathcal{O}\left(\frac{d\sigma}{\alpha^{1/2}\epsilon^{1/2}}\right)$ by Theorem 2.4. Hence, the total number of calls to the stochastic evaluation oracle is $\tilde{\mathcal{O}}\left(\frac{d^2 L^2 \sigma}{\alpha^{5/2}\epsilon^{3/2}}\right)$. Similarly, under Assumptions 2.1 and 2.5 the number of calls to stochastic evaluation at each iteration is $\mathcal{O}\left(\frac{d^{1/2}\sigma}{\alpha^{1/2}\epsilon^{1/2}}\right)$ by Theorem 2.7. Hence, the total number of calls to stochastic evaluation oracle is $\tilde{\mathcal{O}}\left(\frac{d^{3/2} L^2 \sigma}{\alpha^{5/2}\epsilon^{3/2}}\right)$. $\square$

## D  PROOFS FOR GRADIENT ESTIMATION

**Proposition 2.3.** *Let $X \in \mathbb{R}$ be a random variable such that $\mathbb{E}\|X - \mathbb{E}[X]\|^2 \leq \sigma^2$. Given two reals $t \geq 0$ and $\epsilon \in (0, 1)$, then there is a unitary operator $P_{t,\epsilon}^X : |0\rangle |0\rangle \mapsto |\phi_X\rangle |0\rangle$ acting on $\mathcal{H}_X \otimes \mathcal{H}_{aux}$*

*that can be implemented using $\tilde{\mathcal{O}}(t\sigma \log(1/\epsilon))$ quantum experiments and binary oracle queries to $X$ such that*

$$\| |\phi_X\rangle - e^{it\mathbb{E}[X]} |0\rangle \| \leq \epsilon,$$

*with probability at least $8/9$.*

*Proof.* The proof constructs a sequence of unitary operators using the binary-to-phase conversion algorithm for different quantiles of $X$. We begin by randomly drawing a classical sample $s$ from the distribution that generates $X$. By Chebyshev's inequality,

$$\Pr[|s - \mathbb{E}[X]|] \geq 3\sigma] \leq \frac{1}{9}. \tag{76}$$

We consider the case $|s - \mathbb{E}[X]|$ is smaller than $3\sigma$ which holds with probability $8/9$. Next, we define the random variable $Y = X - s$. Additionally, we introduce a random variable $Y_{a,b}$, a truncated version of $Y$, where values of $Y$ outside the interval $[a, b)$ are set to zero. The expectation $\mathbb{E}[Y_{0,\infty}]$ can be expressed as a sum:

$$\mathbb{E}[Y_{0,\infty}] = \mathbb{E}[Y_{0,1}] + \sum_{k=1}^{K} 2^k \mathbb{E}\left[\frac{Y_{2^{k-1}, 2^k}}{2^k}\right] + \mathbb{E}[Y_{2^K, \infty}]. \tag{77}$$

We define the unitary operator $P_{t,\epsilon}^{Y_{a,b}}$, which implements the phase oracle for $\mathbb{E}[Y_{a,b}]$ with an error of at most $\epsilon$. The unitary $P_{t,\epsilon/2}^{Y_{0,\infty}}$ can be implemented as the following product:

$$P_{t,\epsilon/2}^{Y_{0,\infty}} = P_{t,\epsilon/6}^{Y_{0,1}} \left(\prod_{k=1}^{K} P_{t,\epsilon/6K}^{Y_{2^{k-1}, 2^k}}\right) P_{t,\epsilon/6}^{Y_{2^K, \infty}}. \tag{78}$$

When $K = \log\left(\frac{120\sigma^2 t}{\epsilon}\right)$, the operator $P_{t,\epsilon/6}^{Y_{2^K, \infty}}$ is effectively the identity operator, as:

$$\left| |0\rangle - e^{it\mathbb{E}[Y_{2^K, \infty}]} |0\rangle \right| \leq t\mathbb{E}[Y_{2^K, \infty}] \leq \frac{\epsilon}{6}. \tag{79}$$

The last inequality holds because:

$$\mathbb{E}[Y_{2^K, \infty}] = \sum_{Y \geq 2^K} \Pr(Y)Y \leq \sum_{Y} \frac{1}{2^K} \Pr(Y)Y^2 = \frac{\mathbb{E}\|Y\|^2}{2^K} \tag{80}$$

$$\leq \frac{2\mathbb{E}\|X - \mathbb{E}[X]\|^2 + 2\|s - \mathbb{E}[X]\|^2}{2^K} \tag{81}$$

$$\leq \frac{20\sigma^2}{2^K} = \frac{\epsilon}{6t}, \tag{82}$$

where the inequality in the second line follows from the definition of $Y$ and Young's inequality. Since $X_{0,1}$ is bounded between 0 and 1, we can implement $P_{t,\epsilon/6}^{Y_{0,1}}$ using $\tilde{\mathcal{O}}(1)$ queries to $X$ via the binary-to-phase conversion algorithm (Lemma 2.12 in Cornelissen et al. (2022)). We need to show how to implement $P_{t,\epsilon/6K}^{Y_{a,b}}$ when $b > 1$. We start by defining the unitary operator:

$$V_{a,b} : |0\rangle |0\rangle \mapsto \sum_{Y} \sqrt{\Pr(Y)} |Y_{a,b}/b\rangle |0\rangle \tag{83}$$

$$\mapsto \sum_{Y} \sqrt{\Pr(Y)} |Y_{a,b}/b\rangle \left(\sqrt{Y_{a,b}/b} |0\rangle + \sqrt{1 - Y_{a,b}/b} |1\rangle\right) \tag{84}$$

$$= \sqrt{\mathbb{E}[Y_{a,b}/b]} |\psi_0\rangle |0\rangle + \sqrt{1 - \mathbb{E}[Y_{a,b}/b]} |\psi_1\rangle |1\rangle, \tag{85}$$

where the $|\psi_0\rangle$ and $|\psi_1\rangle$ are normalized quantum states. Noting that

$$\mathbb{E}[Y_{a,b}/b] \leq \frac{1}{b} \sum_{a \leq Y \leq b} \Pr(Y)Y \leq \frac{1}{ab} \sum_{a \leq Y \leq b} \Pr(Y)Y^2 \tag{86}$$

$$= \frac{1}{ab} \mathbb{E}\|Y\|^2 \leq \frac{\sigma^2}{ab}, \tag{87}$$

we can apply the linear amplitude amplification algorithm (see (Cornelissen et al., 2022, Proposition 2.10)) to implement the operator:

$$W_{a,b} : |0\rangle |0\rangle \mapsto \sqrt{p_{a,b}} |\psi_0\rangle |0\rangle + \sqrt{1 - p_{a,b}} |\psi_1\rangle |1\rangle , \tag{88}$$

such that,

$$\left| \sqrt{p_{a,b}} - \sqrt{\frac{\mathbb{E}[Y_{a,b}/b]}{\sigma^2/(ab)}} \right| \leq \frac{\epsilon}{24Ktb} \tag{89}$$

using $\tilde{\mathcal{O}}(\sqrt{ab}/\sigma)$ calls to $V_{a,b}$. Let $t' = t\sigma^2/a$. Using the binary-to-phase conversion algorithm, we then implement $|\phi_{a,b}\rangle = e^{it\mathbb{E}[Y_{a,b}]} |0\rangle$ with $\tilde{\mathcal{O}}(t\sigma^2/a)$ calls to $W_{a,b}$ up to an operator norm error of at most $\frac{\epsilon}{12K}$. By using the triangular inequality,

$$\|W_{a,b} |0\rangle - e^{it\mathbb{E}[Y_{a,b}]} |0\rangle \| = \|e^{it' p_{a,b}} |0\rangle - e^{it\mathbb{E}[Y_{a,b}]} |0\rangle \| \tag{90}$$

$$\leq t' \left| p_{a,b} - \frac{\mathbb{E}[Y_{a,b}/b]}{\sigma^2/(ab)} \right| + \frac{\epsilon}{12K} \tag{91}$$

$$\leq 2t' \left| \sqrt{p_{a,b}} - \sqrt{\frac{\mathbb{E}[Y_{a,b}/b]}{\sigma^2/(ab)}} \right| + \frac{\epsilon}{12K} \tag{92}$$

$$\leq \frac{\epsilon}{6K}. \tag{93}$$

Thus, the total implementation of $P_{t,\epsilon/6K}^{Y_{a,b}}$ requires $\tilde{\mathcal{O}}(t\sigma\sqrt{a/b})$ calls to $V_{a,b}$. This implies that each term in the product can be implemented using $\tilde{\mathcal{O}}(t\sigma)$ quantum experiments and binary query oracles to $Y$. Finally, we apply the phase $e^{its}$ to the resulting state to implement $P_{t,\epsilon/2}^{X_{0,\infty}}$. Similarly, we use the same method to implement $P_{t,\epsilon/2}^{X_{-\infty,0}}$, and take the product:

$$P_{t,\epsilon}^X = P_{t,\epsilon/2}^{X_{0,\infty}} P_{t,\epsilon/2}^{X_{-\infty,0}}. \tag{94}$$

This concludes the proof. $\qquad \square$

**Lemma D.1.** *Suppose we run Algorithm 4 with the phase oracle in Proposition 2.3 with evaluation accuracy $\epsilon' = \frac{\epsilon^2}{d^2\beta}$ to $f(\mathbf{x}, \xi)$. Let $\tilde{\mathbf{g}}$ denote the output. Then, under Assumptions 2.1 and 2.2,*

$$\|\tilde{\mathbf{g}} - \nabla f(\mathbf{x})\| \leq \epsilon,$$

*with probability at least $5/9$ using $\tilde{\mathcal{O}}(\frac{\sigma d}{\epsilon})$ queries to $f(\mathbf{x}; \xi)$.*

*Proof.* To be able to run the quantum gradient estimation algorithm, we need to implement $O_F$ that maps

$$O_F |\mathbf{x}\rangle \mapsto e^{i\mathbb{E}_\xi[F(\mathbf{x},\xi)]} |\mathbf{x}\rangle , \tag{95}$$

where $F(\mathbf{x}; \xi) = \frac{N}{2Ll}(f(\mathbf{x}_0 + \frac{l}{N}(\mathbf{x} - N/2); \xi) - f(\mathbf{x}_0; \xi))$. Let $\mathbf{y} = \frac{l}{N}(\mathbf{x} - N/2)$, the variance of $F(\mathbf{x}, \xi)$ is

$$\mathbb{E}\|F(\mathbf{x}; \xi) - \mathbb{E}[F(\mathbf{x}; \xi)]\|^2 = \mathbb{E} \left\| \int_0^1 \langle \nabla f(\mathbf{x} + t\mathbf{y}; \xi) - \nabla f(\mathbf{x} + t\mathbf{y}), \mathbf{y} \rangle dt \right\|^2 \tag{96}$$

$$\leq \|y\|^2 \int_0^1 \mathbb{E}\|\nabla f(\mathbf{x} + t\mathbf{y}; \xi) - \nabla f(\mathbf{x} + t\mathbf{y})\|^2 dt \tag{97}$$

$$\leq \sigma^2 l^2 d. \tag{98}$$

Hence, implementing $e^{i\mathbb{E}[F(\mathbf{x},\xi)]}$ takes $\tilde{\mathcal{O}}(\sigma l d^{1/2} \frac{N}{Ll}) = \tilde{\mathcal{O}}(\frac{\sigma}{\epsilon'^{1/2}\beta^{1/2}}) = \tilde{\mathcal{O}}(\frac{\sigma d}{\epsilon})$ queries to stochastic zeroth-order oracle and succeeds with probability $8/9$. Since Algorithm 4 uses $\tilde{\mathcal{O}}(1)$ queries to $O_F$ by Lemma A.4 and succeeds with probability $2/3$, the total query complexity is $\tilde{\mathcal{O}}(\frac{\sigma d}{\epsilon})$ and success probability is at least $5/9$ due to union bound. $\qquad \square$

**Theorem 2.4.** *Suppose that the potential function $f$ satisfies Assumptions 2.1 and 2.2 and further suppose that $\|\nabla f(\mathbf{x})\| \leq M^3$ for all $\mathbf{x}$. Then, given a real $\hat{\sigma} > 0$, there exists a quantum algorithm that outputs a random vector $\mathbf{g}$ such that*

$$\mathbb{E}[\mathbf{g}] = \nabla f(\mathbf{x}), \quad and \quad \mathbb{E}\|\mathbf{g} - \nabla f(\mathbf{x})\|^2 \leq \hat{\sigma}^2$$

*using $\tilde{O}(\frac{\sigma d}{\hat{\sigma}})$ queries to the stochastic evaluation oracle.*

*Proof.* Suppose that we run Algorithm 4 in Lemma D.1 $T$ times with target accuracy $\frac{\hat{\sigma}}{2}$, then compute the median (coordinate-wise) of these outputs. If the result has norm smaller than $M$, we output this vector. Otherwise, we output all 0 vector. Let $\mathbf{v}$ be the output of this algorithm. Since the algorithm in Lemma D.1 outputs a vector $\tilde{\mathbf{g}}$ such that $\|\tilde{\mathbf{g}} - \nabla f(\mathbf{x})\| \leq \frac{\hat{\sigma}}{2}$ with high probability, then by Chernoff bound and union bound over each dimension, at least $\frac{T}{2}$ of the outputs satisfy $\|\tilde{\mathbf{g}} - \nabla f(\mathbf{x})\| \leq \hat{\sigma}$ with probability at least $1 - 2\exp(-T^2/24)$. Since the norm of the gradient is $M$, when the condition fails, the error is $\|\tilde{\mathbf{g}} - \nabla f(\mathbf{x})\| \leq M$. Then in expectation,

$$\mathbb{E}\|\mathbf{v} - \nabla f(\mathbf{x})\|^2 \leq \frac{\hat{\sigma}^2}{4} + 2\exp(-T^2/24)M^2. \tag{99}$$

Setting $T^2 = 24 \log\left(\frac{8M^2}{3\hat{\sigma}^2}\right)$ gives $\mathbb{E}\|\mathbf{v} - \nabla f(\mathbf{x})\|^2 \leq \hat{\sigma}^2$. Hence, the overhead to Lemma D.1 to make the output smooth is at most logarithmic. Finally, we can use this algorithm as the biased stochastic gradient estimator in Algorithm 3 and obtain an unbiased estimator $\mathbf{g}$. $\square$

**Lemma 2.6.** *Under Assumptions 2.1 and 2.5, Algorithm 1 returns a vector $\mathbf{v}$ such that*

$$\|\mathbf{v} - \nabla f(\mathbf{x})\| \leq \epsilon \tag{13}$$

*with high probability using $\tilde{\mathcal{O}}(\sigma d^{1/2}\epsilon^{-1})$ queries to the stochastic evaluation oracle.*

*Proof.* As the algorithm essentially computes the expectation of $\mathbb{E}_\xi[\tilde{\mathbf{g}}(\mathbf{x}, \xi)]$, we need to prove that $\mathbb{E}_\xi[\tilde{\mathbf{g}}(\mathbf{x}, \xi)]$ is close to $\nabla f(\mathbf{x})$. We consider the case that Algorithm 4 returns $\epsilon/8$ accurate estimate whenever the function $f$ behaves like $\beta$ smooth inside the grid points. Furthermore, we consider the case $\|\mathbf{s} - \nabla f(\mathbf{x})\| \leq 2\sigma$. Both conditions are in fact achieved with high probability. Let $S \subseteq \Xi$ be a set such that the output of quantum gradient estimation (Algorithm 4) $\mathbf{g}$ satisfies $\|\mathbf{g} - \nabla f(\mathbf{x}, \xi)\| \leq \frac{\epsilon}{8}$. Let $S' = \Xi - S$. We can consider the difference in $L_2$ norm separately for $S$ and $S'$ using triangular inequality.

$$\|\mathbb{E}_\xi\tilde{\mathbf{g}}(\mathbf{x}, \xi) - \nabla f(\mathbf{x})\| \leq \|\mathbb{E}_S(\tilde{\mathbf{g}}(\mathbf{x}, \xi) - \nabla f(\mathbf{x};\xi))\| + \|\mathbb{E}_{S'}(\tilde{\mathbf{g}}(\mathbf{x}, \xi) - \nabla f(\mathbf{x};\xi))\|. \tag{100}$$

We first analyze the first term. The contribution to the first term is either due to gradient estimation error $\frac{\epsilon}{8}$ or it is due to the fact that $\mathbf{g}$ is replaced by $\mathbf{s}$ because $\|\mathbf{g} - \mathbf{s}\| > D$. Suppose that $S_1 = \{\xi \in \Xi : \|\mathbf{g}(\mathbf{x};\xi) - s\| \leq D\}$ and $S_2 = S - S_1$. We can separate the error further for both cases using triangular inequality.

$$\|\mathbb{E}_S(\tilde{\mathbf{g}}(\mathbf{x}, \xi) - \nabla f(\mathbf{x};\xi))\| \leq \mathbb{E}_{\xi \in S_1}\|(\mathbf{g}(\mathbf{x}, \xi) - \nabla f(\mathbf{x}, \xi))\| + \mathbb{E}_{\xi \in S_2}\|(\mathbf{s} - \nabla f(\mathbf{x}, \xi))\| \tag{101}$$

$$\leq \mathbb{E}_{\xi \in S_1}\|(\mathbf{g}(\mathbf{x}, \xi) - \nabla f(\mathbf{x}, \xi))\| + \mathbb{E}_{\xi \in S_2}\|(\mathbf{s} - \mathbf{g}(\mathbf{x};\xi))\| \tag{102}$$

$$+ \mathbb{E}_{\xi \in S_2}\|(\mathbf{g}(\mathbf{x};\xi) - \nabla f(\mathbf{x}, \xi))\| \tag{103}$$

$$\leq \frac{\epsilon}{8} + \frac{\mathbb{E}\|\mathbf{s} - \nabla f(\mathbf{x}, \xi))\|^2}{D} + \frac{\epsilon}{8}. \tag{104}$$

The first inequality is due to the fact that for any $\xi \in S_2$, Algorithm 1 replaces $\mathbf{g}$ by $\mathbf{s}$. The last inequality follows from the fact that $\|\mathbf{g}(\mathbf{x};\xi) - \nabla f(\mathbf{x};\xi)\| \leq \frac{\epsilon}{8}$ for any $\xi \in S$ and $\mathbb{E}_{\xi \in S_2}\|\mathbf{s} - \mathbf{g}(\mathbf{x};\xi)\| \leq \frac{\mathbb{E}\|\mathbf{s}-\mathbf{g}(\mathbf{x};\xi)\|^2}{D}$ since for any $\xi \in S_2$ we have $\|\mathbf{g}(\mathbf{x};\xi) - \mathbf{s}\| > D$. As $\|\mathbf{s} - \nabla f(\mathbf{x})\| \leq 2\sigma$,

$$\mathbb{E}\|\mathbf{s} - \mathbf{g}(\mathbf{x};\xi)\|^2 \leq 2\mathbb{E}\|\mathbf{s} - \nabla f(\mathbf{x};\xi)\|^2 + 2\mathbb{E}\|\nabla f(\mathbf{x};\xi) - \mathbf{g}(\mathbf{x};\xi)\|^2 \tag{105}$$

$$\leq 10\sigma^2. \tag{106}$$

---

[3]One can show that the norm of the gradient is bounded by a function of problem parameters throughout the trajectory of HMC or LMC due to smoothness. Since the dependency on $M$ is logarithmic, we do not give an explicit bound on $M$.

Then, for $D = \frac{40\sigma^2}{\epsilon}$, we have $\frac{\mathbb{E}\|\mathbf{s} - \mathbf{g}(\mathbf{x};\xi)\|^2}{D} \leq \frac{\epsilon}{4}$. Therefore, $\|\mathbb{E}_S(\tilde{\mathbf{g}}(\mathbf{x},\xi) - \nabla f(\mathbf{x};\xi))\| \leq \frac{\epsilon}{2}$.

The term due to $S'$ comes from the case where gradient estimation fails. Notice that whenever gradient estimation fails, we have $\|\tilde{\mathbf{g}}(\mathbf{x};\xi) - \nabla f(\mathbf{x})\| \leq \max(D, 2\sigma)$. Gradient estimation only fails when $f(\mathbf{x};\xi)$ has smoothness constant larger than $\beta$. Using Markov's inequality this happens with probability at most $\frac{L}{\beta}$. Then,

$$\mathbb{E}_{S'}\|(\tilde{\mathbf{g}}(\mathbf{x},\xi) - \nabla f(\mathbf{x}))\| \leq \frac{L}{\beta}\max(D, 2\sigma) \leq \frac{\epsilon}{4} \tag{107}$$

for $\beta = \frac{160L\sigma^2}{\epsilon^2}$ and $\sigma \geq \epsilon$. This implies that non-smooth branches do not affect the expectation by replacing $\mathbf{g}$ with $\tilde{\mathbf{g}}$. Furthermore, the variance of $\tilde{\mathbf{g}}(\mathbf{x})$ is

$$\mathbb{E}_\xi\|\tilde{\mathbf{g}}(\mathbf{x},\xi) - \mathbb{E}[\tilde{\mathbf{g}}(\mathbf{x},\xi)]\|^2 \leq 2\mathbb{E}\|\tilde{\mathbf{g}}(\mathbf{x},\xi) - \nabla f(x)\|^2 + 2\|\mathbb{E}[\tilde{\mathbf{g}}(\mathbf{x},\xi)] - \nabla f(x)\|^2 \tag{108}$$

$$\leq 2\mathbb{E}_{S'}\|\tilde{\mathbf{g}}(\mathbf{x},\xi) - \nabla f(x)\|^2 + 2\mathbb{E}_S\|\tilde{\mathbf{g}}(\mathbf{x},\xi) - \nabla f(x)\|^2 + 2\epsilon^2 \tag{109}$$

$$\leq \frac{2Ld}{\beta}\max(D^2, 4\sigma^2) + 2\mathbb{E}\|\nabla f(\mathbf{x};\xi) - \nabla f(\mathbf{x})\|^2 + 2\mathbb{E}\|\mathbf{s} - \nabla f(\mathbf{x})\|^2 + 3\epsilon^2 \tag{110}$$

$$= \mathcal{O}(\sigma^2). \tag{111}$$

Therefore we can use quantum mean estimation to output $\epsilon$ accurate vector $\boldsymbol{v}$ such that $\|\boldsymbol{v} - \nabla f(\mathbf{x})\| \leq \epsilon$ using $\tilde{\mathcal{O}}(\sigma d^{1/2}/\epsilon)$ calls to algorithm $\mathcal{A}$. Since algorithm $\mathcal{A}$ uses $\tilde{\mathcal{O}}(1)$ queries to evaluation oracle, total stochastic evaluation complexity is $\tilde{\mathcal{O}}(\sigma d^{1/2}/\epsilon)$. $\qquad\square$

**Theorem 2.7** (Smooth Gradient). *Suppose that the potential function $f$ satisfies Assumptions 2.1 and 2.5 and further suppose that $\|\nabla f(\mathbf{x})\| \leq M$ for all $\mathbf{x}$. Then, given a real $\hat{\sigma} > 0$, there exists a quantum algorithm that outputs a random vector $\mathbf{g}$ such that*

$$\mathbb{E}[\mathbf{g}] = \nabla f(\mathbf{x}), \quad and \quad \mathbb{E}\|\mathbf{g} - \nabla f(\mathbf{x})\|^2 \leq \hat{\sigma}^2 \tag{14}$$

*using $\tilde{\mathcal{O}}(\frac{\sigma d^{1/2}}{\hat{\sigma}})$ queries to the stochastic evaluation oracle in expectation.*

*Proof.* Suppose that we run Algorithm 1 $T$ times with target accuracy $\frac{\hat{\sigma}}{2}$, then compute the median (coordinate-wise) of these outputs. If the result has norm smaller than $M$, we output this vector. Otherwise, we output all $0$ vector. Let $\mathbf{v}$ be the output of this algorithm. Since Algorithm 1 outputs a gradient $\mathbf{v}$ such that $\|\mathbf{v} - \nabla f(\mathbf{x})\| \leq \hat{\sigma}/2$ with high probability (say $2/3$), then by Chernoff bound and union bound over each dimension, at least $\frac{T}{2}$ of the outputs satisfy $\|\mathbf{v} - \nabla f(\mathbf{x})\| \leq \hat{\sigma}$ with probability at least $1 - 2\exp(-T^2/24)$. Since the norm of the gradient is $M$, when the condition fails the error is $\|\mathbf{v} - \nabla f(\mathbf{x})\| \leq M$. Then in expectation,

$$\mathbb{E}\|\mathbf{v} - \nabla f(\mathbf{x})\|^2 \leq \frac{\hat{\sigma}^2}{4} + 2\exp(-T^2/24)M^2. \tag{112}$$

Setting $T^2 = 24\log\left(\frac{8M^2}{3\hat{\sigma}^2}\right)$ gives $\mathbb{E}\|\mathbf{v} - \nabla f(\mathbf{x})\|^2 \leq \hat{\sigma}^2$. Hence, the overhead is at most logarithmic. Finally, we run Algorithm 3 to obtain an unbiased estimator $\mathbf{g}$. $\qquad\square$

# E   PROOFS FOR OPTIMIZATION

To be able to characterize the run-time of the algorithm, we first need to characterize the Log-Sobolev constant of $f_v$. To achieve this, we use the following LSI perturbation lemma by Holley-Stroock Holley & Stroock (1987).

**Lemma E.1.** *Let $\rho$ be the Log-Sobolev constant of the Gibbs distribution with potential $F$. Then, the Log-Sobolev constant of $f$ satisfies,*

$$\alpha \geq \rho e^{-|\sup_x(f(x) - F(x)) - \inf_x(f(x) - F(x))|}. \tag{113}$$

Next we present the proofs of Lemma 4.4 and Theorem 4.5.

**Lemma 4.4.** *Let $\pi_v^\beta = \frac{e^{-\beta f_v(\mathbf{x})}}{\int e^{-\beta f_v(\mathbf{x})}d\mathbf{x}}$. If $\beta = \Theta(d/\epsilon)$ and $v \leq \mathcal{O}(\frac{\epsilon}{Md})$, then sampling from $\pi_v^\beta$ returns $\epsilon$ approximate optimizer for $f$ with probability at least 0.1.*

*Proof.* Consider the following quantity that quantifies the optimization error with respect to original distribution $\pi^\beta$.

$$\varepsilon(\beta) = \mathbb{E}_{\pi^\beta}[f(\mathbf{x})] \tag{114}$$

which can be written explicitly,

$$\varepsilon(\beta) = \frac{\int_{\mathbb{R}^d} f(\mathbf{x})\exp(-\beta f(\mathbf{x}))d\mathbf{x}}{\int_{\mathbb{R}^d}\exp(-\beta f(\mathbf{x}))d\mathbf{x}} \tag{115}$$

Consider the derivative of $\varepsilon(\beta)$ with respect to $\beta$.

$$\frac{d\varepsilon(\beta)}{d\beta} = \frac{\beta\left(\int_{\mathbb{R}^d} f(\mathbf{x})\exp(-\beta f(\mathbf{x}))d\mathbf{x}\right)^2 - \beta\left(\int_{\mathbb{R}^d} f(\mathbf{x})\exp(-\beta f^2(\mathbf{x}))d\mathbf{x}\right)\left(\int_{\mathbb{R}^d}\exp(-\beta f(\mathbf{x}))\right)d\mathbf{x}}{\left(\int_{\mathbb{R}^d}\exp(-\beta f(\mathbf{x}))d\mathbf{x}\right)^2} \tag{116}$$

$$= -\mathrm{Var}(\beta^{1/2}f(\mathbf{x}))) \tag{117}$$

Since variance is positive quantity, $\frac{d\varepsilon(\beta)}{d\beta}$ is negative and its absolute value is bounded by

$$\left|\frac{d\varepsilon(\beta)}{d\beta}\right| \leq \mathrm{Var}(\beta^{1/2}(f(\mathbf{x}))) \leq M\beta\mathrm{Var}(\mathbf{x}) \leq \mathcal{O}(d) \tag{118}$$

where the last inequality is due to the fact that tails of $\pi^\beta$ is upper bounded by a Gaussian with variance $\Omega(1/\beta)$. Since $\varepsilon(\infty) = 0$, we can write

$$\varepsilon(\beta) = \lim_{\beta_u \to \infty} \int_{\beta_u}^{\beta} \frac{d\varepsilon(\beta)}{d\beta}d\beta \leq \mathcal{O}(d/\beta) \tag{119}$$

Let $f(x^\star) = \min_{\mathbf{x}} f = 0$ without loss of generality. Then, for $\beta > \Theta(d/\epsilon)$, $\mathbb{E}_{\pi^\beta}[f(\mathbf{x}) - f(\mathbf{x}^\star)] = \varepsilon(\beta) \leq \epsilon$. By Markov's inequality

$$\Pr[f(\mathbf{x}) - f(\mathbf{x}^\star) \geq \epsilon)] \leq 0.05 \tag{120}$$

For $\nu \leq \mathcal{O}(\frac{\epsilon}{Md})$,

$$\beta(f_\nu(\mathbf{x}) - f(\mathbf{x})) \leq \beta M\nu \leq \mathcal{O}(1) \tag{121}$$

Therefore, the total variation distance between $\pi^\beta$ and $\pi_\nu^\beta$ can be made smaller than 0.05 by scaling $\nu$ by a constant. Then,

$$\Pr[f(\mathbf{x}) - f(\mathbf{x}^\star) \geq \epsilon)] \leq 0.1 \tag{122}$$

This concludes the proof.

$\square$

**Theorem 4.5.** *Suppose that $f$ satisfies Assumptions 4.1 and 4.2. Then, there exists a quantum algorithm that returns $\epsilon$ approximate minimizer for $f$ with probability at least 0.1 using $\tilde{\mathcal{O}}\left(\frac{d^{9/2}}{\epsilon^{3/2}}\right)$ queries to the stochastic evaluation oracle for $f$.*

*Proof.* We consider the potential function $\beta f_v(x)$ where $\beta$ is the inverse temperature parameter. By Lemma 4.4, sampling from $\pi_v^\beta \propto e^{-\beta f_v}$ returns $\frac{\epsilon}{2}$ approximate minimizer for $f$ with high probability (say 0.9) for sufficiently large $\beta = \mathcal{O}(\frac{d}{\epsilon})$. Suppose that we sample from a probability distribution $\mu$ such that

$$\mathrm{TV}(\mu, \pi_v^\beta) \leq 0.1. \tag{123}$$

Then, the sample must be $\frac{\epsilon}{2}$ minimizer for $f$ with probability at least 0.8. Therefore, it is sufficient to sample from $\pi_v^\beta$ up to a constant TV distance.

We need to characterize the sampling complexity from $\pi_v^\beta$. From Bakry Emery theorem, Log Sobolev constant $\rho$ of $\beta F$ satisfies $\rho \geq \frac{\beta\mu}{2}$ where $\mu$ is the strong convexity constant of $F$. Let

$M' = \max(M, 1)$ and take $v = \frac{\epsilon}{2M'd}$. Then using LSI perturbation result Lemma E.1 by Holley-Strock, we have $\alpha \geq \frac{\beta\mu}{2}e^{-3\beta\epsilon/d} = \Omega(\frac{\mu d}{\epsilon})$. The uniform perturbation holds due to the inequality $|f_v - F| \leq |f_v - f| + |f - F| \leq vM + \frac{\epsilon}{d} \leq \frac{3\epsilon}{2d}$. Since $\beta f_v$ is a smooth function with smoothness constant $L = \mathcal{O}(\frac{\beta M\sqrt{d}}{v}) = \mathcal{O}(\frac{d^{5/2}M^2}{\epsilon^2})$ by Proposition 4.3, the number of calls to stochastic evaluation oracle to sample from $\pi_v$ is $\tilde{\mathcal{O}}(\frac{L^2 d^2}{\alpha^{5/2}}) = \tilde{\mathcal{O}}(\frac{M^4 d^{9/2}}{\mu^{5/2}\epsilon^{3/2}})$ by Theorem 3.4. Hence, we can optimize $f$ in polynomial time. $\qquad\square$

