# OpenReview forum: "Quantum Speedups for Sampling and Non-convex Optimization with Stochastic Zeroth Oracles"
_ICLR.cc/2026/Conference — Submitted to ICLR 2026_

### Official Review · Reviewer_aUSH · 2025-10-29

**Soundness:** 2
**Presentation:** 2
**Contribution:** 2
**Rating:** 2
**Confidence:** 4

**Summary:**

This paper presents a novel quantum algorithmic framework for accelerating sampling and non-convex optimization in a stochastic zeroth-order setting. The theoretical contribution is significant, as it cleverly combines quantum gradient estimation, quantum mean estimation, and classical sampling theory to provide a provable quantum speedup under specific conditions. However, the paper suffers from fundamental issues regarding the justification of its core model and its general applicability, which currently limit the solidity of its claims and the breadth of its impact.

**Strengths:**

This work presents a comprehensive theoretical framework that integrates tools from quantum computing (gradient estimation, mean estimation) and classical numerical analysis (MLMC, convergence theory for samplers) to achieve an end-to-end complexity analysis with polynomial speedups.

**Weaknesses:**

There are some major concerns affecting the score of this paper, as follows:

1. Questionable Fairness and Realism of the Core Oracle Model

The paper's technical approach relies crucially on a strong oracle assumption: the ability to query the stochastic function at two different points using the same random seed (i.e., "reproducible randomness"). This is a fundamentally more powerful model than the standard stochastic zeroth-order oracle used by the classical baselines, which typically allows for independent sampling on each query. Demonstrating a quantum speedup against classical algorithms that operate in a weaker, standard model is arguably unfair. The claimed speedup might be a direct consequence of this stronger assumption rather than a pure algorithmic improvement.

Further, this strong assumption severely restricts the generality of the proposed algorithms. They are primarily applicable to finite-sum problems, where fixing the random seed corresponds to selecting a specific data index. For many important real-world problems (e.g., optimization based on physical experiments, interactions with non-stationary systems), the algorithm is not directly applicable. The paper should more explicitly acknowledge this limitation rather than presenting its results as a general "stochastic zeroth-order" acceleration.

2. Insufficient Analysis of Quantum Resource Costs

While the focus on query complexity is standard for a theoretical paper, the complete omission of other quantum resource costs may mislead readers about the algorithm's practical feasibility. What is the asymptotic scaling of the number of qubits required to construct the phase oracle (Proposition 2.3) and run the robust estimation framework (Algorithm 1)? Is this scaling polynomial in the dimension and the precision? This information is crucial for assessing practical viability.

3. Lack of Comparison with Relevant Quantum Works

The paper chooses to compare its performance against classical zeroth-order algorithms. However, it lacks a critical comparison with relevant quantum algorithms. For the most natural application scenario—finite-sum optimization—there exist other quantum-accelerated methods. How does the proposed sampling-based framework compare to these approaches? Is it superior in terms of query complexity, generality, or implementation difficulty?

**Questions:**

1. The introduction and discussion should clearly state that the work relies on a "strengthened oracle model with reproducible randomness."  Meanwhile, the authors should discuss whether this strong assumption is necessary for achieving the speedup.

2. A fairer comparison would be against a classical algorithm that is also granted the same powerful oracle, or the paper should explicitly frame the speedup as being achieved at the cost of reduced generality.

3. A rough asymptotic analysis of the quantum resource requirements should be provided in the appendix or discussion.

4. A dedicated paragraph in the related work should discuss the anticipated performance of the proposed algorithm against existing quantum optimization/sampling algorithms under the same (finite-sum) setting.

---

> ### Author Response · Authors · 2025-11-18
> **Response to the Concerns of Reviewer aUSH**
>
> We appreciate the reviewer’s feedback. We have carefully examined all of the raised concerns. However, we believe that most of these concerns arise from omission of details already addressed and substantiated in the paper. We also find some of these concerns to be out of scope of our work as we elaborate below.
>
> **Access model**:
> The reviewer raised a concern about our assumption that the stochastic oracle allows access to two evaluations of the function using the same random seed ($f(x,\xi)$ and $f(y,\xi)$). This is a fair point; however:
>
> The baseline classical algorithm to which we compare, Roy et al. (2021), uses exactly the same assumption, and discusses it in detail on Page 6. Thus, both the quantum and classical algorithms compared in our paper operate under the same oracle power. Furthermore, this is the standard two-point stochastic oracle used in the classical zeroth-order literature (e.g., Duchi et al. (2015), Nesterov & Spokoiny (2017), Roy et al. (2021)).  In the one-point evaluation setting, the problem becomes challenging for both classical and quantum algorithms. In fact, Roy et al.(2021) also considered this setting, and the complexity is worse than the two-point setting. Quantumly, the challenge is that we can no longer implement quantum mean estimation but only gradient estimation, which can still give a speedup, but this would not yield an interesting algorithm.
>
> Moreover, in the quantum setting, two-point access is not strictly necessary as long as a superposition over the stochastic distribution can be prepared (which is true for efficiently integrable distributions). In other words, the algorithm does not fundamentally require reproducing the randomness. However, explicitly assuming two-point access avoids technical overhead related to state preparation and ensures a fair comparison with classical methods, since classical algorithms could in principle gain additional power from explicit distributional knowledge. Although it is not clear at the surface level, this assumption is in fact necessary to make a fair comparison to the classical algorithm by Roy et al. (2021).
> Therefore, the speedup does not come from this assumption, but from the inverse Fourier transform implemented in superposition, as described in Section 2.
>
>
> Finally, although the reviewer suggests that the paper presents the results in general “stochastic zeroth-order setting”, this is not true. It is in fact made explicit in multiple locations: in the abstract (lines 16–17), in the main text (Eq. (6), and Lines 209–211), and in Appendix D, where the oracle model is formally defined. We will nevertheless make this assumption even more prominent in the revised version.
>
> **Resource Cost**:
>  The reviewer raises concerns about the resource costs of our algorithms however this was already addressed in the paper.  Our submodules rely on quantum amplitude estimation and the quantum Fourier transform, which are among the most widely used primitives in quantum algorithms (including factoring, counting, search etc.). As stated around lines 344–350, the algorithm requires only polynomially many gates in the dimension $d$ and precision $1/\epsilon$, as our algorithms call these submodules only polynomially many times (Page 5 and Proposition 3). The qubit count also follows directly from the gate complexity and standard constructions in Jordan (2005), Cornelissen et al. (2022), and Chakrabarti et al. (2025). Specificially Algorithm 1 require $ O(\text{poly}(d,\log⁡(1/ϵ)))$ gates and qubits.
>
> **Related Work**:
> Lines 469–485 in page 9 already discuss related work on optimizing approximately convex functions. We believe the reviewer may have overlooked this discussion.  We should also note that our algorithms give improved dimension dependency even though the classical algorithms we compare use more strict assumptions than ours such as sub-Gaussian tails or lower noise levels in the oracle. For the sampling case, we are not aware of any quantum works that operate under the same stochastic zeroth-order assumptions. Prior quantum works—such as Childs et al. (2022)—considered only strongly convex potentials with deterministic access to the evaluation oracle, while Ozgul et al. (2024) studied stochastic gradient access rather than stochastic zeroth-order oracles. In terms of implementation difficulty, our algorithms are significantly easier than these prior works because their algorithms require maintaining coherent quantum state, whereas we only have an hybrid algorithm and do not need to maintain such a quantum state. For finite-sum problems, we are only aware of the results by Zhang et al (2024). However, their result assumes gradient access rather than evaluation access. We are happy to investigate if the reviewer has a specific result for finite-sum potentials in the zeroth-order setting, however we find this to be out of scope of this work because finite sum case was only given as an example that could be application of our work.

---

> > ### Author Response · Authors · 2025-11-18
> > **Response to the Questions of Reviewer aUSH**
> >
> > Although some of these points are already elaborated in above comment, the reviewer can find the responses to the questions below:
> >
> > 1. We will clarify early in the paper that the algorithm relies on a stochastic evaluation oracle with reproducible randomness, consistent with the standard two-point stochastic model.
> > 2. Our comparisons already use the same oracle model as classical algorithms we compare; we will explicitly note this to ensure fairness of comparison.
> > 3. A brief asymptotic discussion of gate and qubit requirements will be added in the appendix, emphasizing polynomial scaling in d and $1/\epsilon$.
> > 4. We will add a short related work paragraph comparing our framework to existing quantum optimization/sampling algorithms under the finite-sum setting. Although we note that the only difference is that one needs to do sub-sampling, and this does not change the contribution of the work significantly.
> >
> >
> > **References**
> >
> > 1. Duchi, J. C., Jordan, M. I., Wainwright, M. J., & Wibisono, A. (2014). Optimal rates for zero-order convex optimization: The power of two function evaluations. arXiv:1312.2139. https://arxiv.org/abs/1312.2139.
> > 2. Nesterov, Y., Spokoiny, V. Random Gradient-Free Minimization of Convex Functions. Found Comput Math 17, 527–566 (2017). https://doi.org/10.1007/s10208-015-9296-2
> > 3. Roy, A., Shen, L., Balasubramanian, K., & Ghadimi, S. (2021). Stochastic Zeroth-order Discretizations of Langevin Diffusions for Bayesian Inference. arXiv:1902.01373. https://arxiv.org/abs/1902.01373
> > 4. Zhang, Y., Zhang, C., Fang, C., Wang, L., & Li, T. (2024). Quantum Algorithms and Lower Bounds for Finite-Sum Optimization. arXiv:2406.03006. https://arxiv.org/abs/2406.03006
> > 5. Childs, A. M., Li, T., Liu, J.-P., Wang, C., & Zhang, R. (2022). Quantum Algorithms for Sampling Log-Concave Distributions and Estimating Normalizing Constants. arXiv:2210.06539. https://arxiv.org/abs/2210.06539
> > 6. Ozgul, G., Li, X., Mahdavi, M., & Wang, C. (2023). Stochastic Quantum Sampling for Non-Logconcave Distributions and Estimating Partition Functions. arXiv:2310.11445. https://arxiv.org/abs/2310.11445

---

> > ### Comment · Reviewer_aUSH · 2025-11-20
> >
> > Thank you for the responses. The author has alleviated my concerns, and I have increased the score.

---

### Official Review · Reviewer_j1qK · 2025-10-31

**Soundness:** 3
**Presentation:** 4
**Contribution:** 3
**Rating:** 8
**Confidence:** 3

**Summary:**

This paper investigates the potential of quantum algorithms to accelerate the optimization of functions with only access to zeroth oracle, in particular nonsmooth and almost convex functions. The authors demonstrate that using quantum mean estimation and jordan's algorithm, it is possible to achieve quadratic speedups over classical methods on various problems. This work derives an algorithm to efficiently and accurately estimate gradients even in the presence of noise and approximation errors. Additionally, this paper shows that the result can be further generalized to non-smooth scenarios via gradient estimation.

**Strengths:**

The algorithm proposed in the paper is explained very well. The paper has a good presentation where it focuses on not only the technical details but also the intuition behind the algorithm. Besides, showing an elegant algorithm for gradient estimation on nonsmooth functions is interesting.

**Weaknesses:**

No significant weekness

**Questions:**

- Is it possible to also prove a lower-bound for the optimization scenario considered in this work?
- Would it be possible to extend the result to some extent to non-convex landscapes?

---

> ### Author Response · Authors · 2025-11-18
> **Response to Reviewer j1qK**
>
> We are glad that the reviewer found the paper both novel and interesting. We are also glad that our paper provides intuitive insights beyond technical details. We thank the reviewer for the positive and supportive feedback.
>
> To address the questions,
>
> 1- As far as we are aware, there are no explicit classical or quantum lower bounds for the exact problems considered in the optimization section. However, the work by Li, Risteski (2016) showed information-theoretic bounds on the noise tolerance level and the algorithmic runtime( d and $\epsilon$ dependency is not explicitly given) for noisy convex optimization problems.
>
> 2- Our sampling results under the log-Sobolev inequality (LSI) allow both convex and non-convex potentials. In fact, LSI is an analogue of PL (Polyak-Lojasiewicz) inequality in optimization, and therefore, the functions can be non-convex.
>
> **References**
>
> 1. Andrej Risteski and Yuanzhi Li. Algorithms and matching lower bounds for approximately-convex optimization. Advances in Neural Information Processing Systems, 29:4752–4760, 2016.

---

### Official Review · Reviewer_wZj3 · 2025-11-01

**Soundness:** 2
**Presentation:** 2
**Contribution:** 3
**Rating:** 4
**Confidence:** 2

**Summary:**

This work proposes quantum algorithms that achieve provable speedups for sampling from Gibbs distributions $\pi \propto e^{-f(x)}$ and for optimizing nonconvex objectives, when only stochastic zeroth-order (function value) access to $f$ is available. The authors introduce new quantum stochastic gradient estimation methods that improve classical query complexities from$\tilde{O}(d^2\sigma^2/\varepsilon^2) $ to $\tilde{O}(d\sigma/\varepsilon)$ or even $ \tilde{O}(d^{1/2}\sigma/\varepsilon)$ under additional smoothness assumptions. These estimators are then used to obtain LMC and HMC with reduced oracle complexity, leading to polynomial quantum speedups in dimension, precision, and noise parameters for both sampling and optimization tasks. The paper further extends these results to nonsmooth and approximately convex optimization, showing that quantum sampling techniques can yield faster convergence in empirical risk minimization–type problems. Theoretical guarantees are established for all algorithms, assuming fault-tolerant quantum computation.

**Strengths:**

- Originality: Introduces the first quantum algorithms achieving provable polynomial speedups for stochastic zeroth-order sampling and optimization, extending quantum gradient estimation to a realistic noisy-oracle model. Provides rigorous convergence and complexity analyses for quantum variants of LMC and HMC, connecting Jordan’s gradient estimation, quantum mean estimation, and MLMC in a novel way.
- Breadth of applicability: Framework covers both strongly convex and nonconvex settings, and further applies to nonsmooth approximately convex optimization, showing broad theoretical relevance.
- Clarity: The paper is well-structured, making the logical flow of ideas easy to follow.
- Significance: Establishes new theoretical baselines for quantum advantages in sampling and optimization, potentially guiding future algorithm design once fault-tolerant quantum hardware becomes available.

**Weaknesses:**

- Proof missing details: could the author explain in section B.1, proof of theorem 3.2, there seems to be a mismatch of $\kappa$ and $\sigma$ in the proof and statement. Could the authors clarify a bit on this?
- Unclear treatment of bounded gradients: Several proofs rely on a global bound $\|\nabla f(x)\|\le M$ without establishing or bounding M in terms of problem parameters. Could the authors clarify a bit on this?
- Novelty of their techniques: much of the techniques of quantum speedups seems to come from quantum gradient estimation and mean estimation. Could the authors explain more about their technical novelty?

**Questions:**

- See the weaknesses part.
- Line 1372: "where the last inequality is due to the fact that tails of $\pi^{\beta}$ is upper bounded by a Gaussian with variance $\Omega(1/\beta)$." could the authors explain more about why this holds, especially relying on what kind of assumptions? It seems not immediately clear to me.

---

> ### Author Response · Authors · 2025-11-18
> **Response to Reviewer wZj3**
>
> We thank the reviewer for the thoughtful feedback. We are pleased that the reviewer recognized both the significance of our contributions. Below we clarify the main points.
>
> **1. Clarification on the Step-Size Conditions in Theorem 3.2**:
> The confusion likely arises because two different sets of conditions appear in the proof of Theorem 3.2. We now clarify their roles:
> In the proof, we restate Theorem B.1 and explicitly write down the conditions required for that theorem to apply. These ensure the validity of the underlying Wasserstein error bound. We then derive additional constraints on the step size and noise level that ensure the two error terms $\Gamma_1$ and $\Gamma_2$ are sufficiently small. These constraints are separate from and complementary to the ones in Theorem B.1. We will make this distinction clearer in the revised version.
>
> **2. On the Bounded Gradient Assumption**:
> The bound on the norm of the gradient is a very valid concern; however, in sampling problems, assuming the norm of the gradient is bounded by a function of $\text{poly}(d)$ is almost equivalent to allowing unbounded gradients, because when the function is smooth, the algorithm almost never encounters a gradient with norm larger than $\text{poly}(d)$. This is somewhat standard in sampling literature. To justify this, let us give a semi-rigorous proof.
>
> Since the function $f$ is smooth, its gradient is Lipshitz, i.e., $\|\nabla f(x)-f(y)\|\leq L \|x-y\|$. Suppose that the function $f$ is minimized at $x^{\star}$ and therefore $\nabla f(x^{\star})=0$. This implies that $\|\nabla f(x)\|\leq L\|x\|$. Of course, since the bound is infinite, this does not give a bounded gradient.
>
> Now, the trick is to notice that both strong log-concavity and LSI imply that the distribution has fast-decaying tails (sub-Gaussian tails in particular). Therefore, the distribution has smaller than $\epsilon$ probability when $\|x-x^{\star}\|$ is large ($\text{poly}(d)$). Hence, we can truncate the space at sufficiently large $R = \text{poly}(d, 1/\epsilon)$ to make sure that the truncated distribution is $\epsilon$-close to the original distribution and run the sampling algorithm on the bounded domain (the boundaries can easily be handled by rejecting the steps that go out of bounds). Therefore, this algorithmic change makes sure that the gradients are always bounded in the trajectory of the algorithm, whereas the sampling error is very small. This algorithmic trick is also employed by other works such as  Zou et al (2019).
> The reason we added this assumption is that the algorithm becomes very simple to explain and avoids the discussion above. Since the final complexity only depends logarithmically on the norm of the gradient, this is a very mild technical assumption, but it improves the presentation quite a bit. We will add more details about this in the revised version.
>
> **3. Novelty of the Quantum Gradient Estimation Components**
> While it is true that the improvements are due to quantum mean estimation and gradient estimation, we do not utilize these tools as off-the-shelf algorithms because quantum gradient estimation does not work in a stochastic bounded variance setting. Our contribution is to propose new quantum gradient estimation algorithms in Section 2. We believe these algorithms are general-purpose and have not appeared elsewhere previously. To summarize the technical algorithmic steps:
>
> 1- In section 2.1, we designed a quantum gradient estimation algorithm by showing that the required quantum state before the quantum Fourier transform can be prepared even though the access to the function is not very accurate, which is the main shortcoming of existing algorithms. This is done by considering different quantiles of the random variable described in section D.
>
> 2- In section 2.2, we also gave another quantum algorithm for gradient estimation in settings where the noisy gradient is not Lipschitz even though the true gradient is Lipschitz. This is a common scenario in machine learning problems; however, it has not been addressed before.
>
> 3- Incorporating these tools requires that one chooses the required precision in the gradient estimation. By optimizing this precision level for different cases, we achieved the claimed speedups. Without optimizing the precision, it is still not clear whether quantum algorithms can yield the claimed speedups because, in certain cases, the error is dominated by the stochasticity of the process itself, not the gradient. Therefore, our speedups are not due to trivial application of existing quantum algorithms.
>
> **References**
> 1. Difan Zou, Pan Xu, and Quanquan Gu. Faster convergence of stochastic gradient Langevin dynamics for non-log-concave sampling. arXiv preprint arXiv:2010.09597, 2021.

---

> > ### Author Response · Authors · 2025-11-18
> > **Response to Question about Tail Bound**
> >
> > In the setting we consider, the function we are tying to optimize is uniformly close to a strongly convex distribution and therefore, the distribution has faster decaying tails than a Gaussian. This can also be showed by showing that the distribution satisfies log-Sobolev inequality (as we have done in proof of Theorem 4.5) and use the well known fact that log-Sobolev inequality implies sub-Gaussian tails.
> >
> > We will add this detail to the proof in the revised version. We thank the reviewer for raising this issue.

---

### Official Review · Reviewer_WWpT · 2025-11-09

**Soundness:** 3
**Presentation:** 3
**Contribution:** 3
**Rating:** 6
**Confidence:** 4

**Summary:**

This paper presents a novel quantum algorithm for stochastic gradient estimation under various smoothness assumptions, leading to quadratic speedups for smooth potential functions. By leveraging this new stochastic gradient estimation subroutine in zeroth-order sampling tasks, this paper proposes two new quantum algorithms that achieve polynomial speedups over existing classical methods. The application of this approach to non-smooth and approximately convex optimization has also been discussed in the paper.

**Strengths:**

- A new quantum gradient estimation subroutine is proposed that overcomes the drawbacks of existing gradient estimation methods. In particular, this paper only requires the expectation value of the Lipschitz constant to be bounded. This is a slightly weaker assumption than many previous results. This is achieved by a careful combination of quantum mean estimation and Jordan's gradient estimation algorithm.
- This quantum gradient estimation subroutine has been applied to both LMC and HMC, and the convergence is analyzed.
- Applications to noisy, approximately convex optimization problems are discussed. This is a prominent problem class with important applications in ML, such as empirical risk minimization.

**Weaknesses:**

- The distance metric ($W_2$) used in Theorem 3.2 appears to be weaker than those in Theorem 3.4. Is this because the analysis of the base classical algorithm (HMC) is less explored compared to LMC? Does the quantum algorithm improve the distance metric?
- The approximate convexity assumption (Assumption 4.1) is very weak in high dimension ($d \gg 1$). Is it possible to relax this assumption further and still obtain quantum speedups? Will quantum algorithms be more competitive in the more "noisy" regime?

**Questions:**

See comments above.

---

> ### Author Response · Authors · 2025-11-18
> **Response to Reviewer WWpT**
>
> We thank the reviewer for both recognizing the contribution of our work and raising questions related to the difference in the distance metrics used in Theorems 3.2 and 3.4 and the noise tolerance in Section 4 regarding nonsmooth optimization. We appreciate the provided feedback.
>
> Here we address these questions/concerns:
>
>  **Distance Metric**:
> Regarding the difference in the distance metrics, it is correct that the Wasserstein distance in Theorem 3.2 is weaker than the distance metric in Theorem 3.4 because Theorem 3.4 derives the result in terms of KL divergence which also implies convergence in terms of TV distance (via Pinsker’s inequality) and $W_2$ distance (via Talagrand’s inequality for distributions that satisfy log-Sobolev inequality.). The main reasons why these theorems choose different approaches are as follows:
> - The result in Theorem 3.4 already accommodates other distance metrics for strongly convex functions because a $\mu$ strongly log concave distribution satisfies LSI inequality with constant $\mu/2$. Therefore, Theorem 3.4 already gives improved bounds for strongly convex functions for the KL and TV distances. However, since Theorem 3.4 derives everything in terms of KL, the convergence rate in terms of W_2 distance is suboptimal.
> - Theorem 3.2 directly analyzes the Wasserstein distance for the Hamiltonian Monte Carlo algorithm and provides improved complexity compared to Theorem 3.4 (See near quadratic improvement in dimension dependency). Indeed, the classical baseline algorithm’s complexity in the stochastic zeroth-order setting is measured by the Wasserstein distance. Since Theorem 3.4 already accounts for other metrics and we don’t have an appropriate classical baseline, the HMC result is only presented in terms of W_2 distance. We also note that both approaches are typically very different, and hence it is typical for a work to choose a certain metric. We compensated by providing results in other metrics in Theorem 3.4. Note that there exist classical results for HMC in terms of the TV distance; however, the oracles are typically full gradient not stochastic evaluation.
>
> **Noise Assumption**:
> For the assumption on the noise level for optimization, we agree that the noise needs to be very weak in high-dimensional problems. The classical works by Li et al. and Belloni et al. also considered the same noise level.
> Unfortunately, this is a theoretical limitation rather than a weakness in the analysis. Risteski et al. in fact showed that a higher noise level cannot be tolerated in general:
>
> **Theorem by Risteski, Andrej & Li, Yuanzhi (2016)**
> For every algorithm $\mathcal{A}$, for every $d, \Delta, \epsilon$ with $\Delta = \tilde{\Omega}(\max(\epsilon^2/\sqrt{d}, \epsilon/d))$, there exists a function $\tilde{f}$ on a convex set in $\mathbb{R}^d$ of diameter $1$, and $f$ is $\delta$ close to an $1$-Lipschitz convex function $f$, such that $\mathcal{A}$ cannot find a point $\tilde{x}$ with $f(\tilde{x})\leq \min_{x}\tilde{f}(x)+\epsilon$ in $\text{poly}(d,1/\epsilon)$ evaluations of $\tilde{f}$.
>
> Hence, by the above theorem, no algorithm can solve this problem by allowing higher noise levels. As this bound is information-theoretical, we expect quantum algorithms to fail when the noise is higher in general. However, this bound on the tolerance level is general, and there might be tradeoffs between the noise level and the query complexity in different settings. A recent work by Chakrabarti et al. investigated the runtime of quantum algorithms by allowing different noise levels. As seen from table 1 in their work, our noise tolerance matches $\epsilon/d$ and is higher than their noise level. Therefore, our noise level seems to saturate the limit. However, lower noise levels can give better query complexities.
>
> Note: The table in Chakrabarti et al. are not for the stochastic setting, so simulated annealing results in the table are not comparable to ours. However, the same papers (Belloni et al and Li et al) also have results in the stochastic setting where we make the comparison in our paper.
>
> References
> 1. Andrej Risteski and Yuanzhi Li. Algorithms and matching lower bounds for approximately-convex optimization. Advances in Neural Information Processing Systems, 29:4752–4760, 2016.
> 2. Shouvanik Chakrabarti, Dylan Herman, Jacob Watkins, Enrico Fontana, Brandon Augustino, Junhyung Lyle Kim, and Marco Pistoia. On speedups for convex optimization via quantum dynamics. arXiv preprint arXiv:2503.24332, 2025.
> 3. Alexandre Belloni, Tengyuan Liang, Hariharan Narayanan, and Alexander Rakhlin. Escaping the local minima via simulated annealing: Optimization of approximately convex functions. In Conference on Learning Theory, pages 240–265. PMLR, 2015.
> 4. Tongyang Li and Ruizhe Zhang. Quantum speedups of optimizing approximately convex functions with applications to logarithmic regret stochastic convex bandits. Advances in Neural Information Processing Systems, 35:3152–3164, 2022.

---

### Meta-Review · Area_Chair_uqmG · 2025-12-31

**Summary:**

The authors study sampling from a Gibbs distributions on a quantum computer. If seen from an optimization perspective, this is related to a subset of convex (and thus non-convex) optimization problems (cf. Assumption 4.1 and https://link.springer.com/article/10.1007/s10107-021-01707-1). All reviewers agree that the proposed combination of quantum mean estimation and Jordan's gradient estimation algorithm is novel and interesting.

All reviewers agree that the assumptions are not spelled out properly, both in terms of the assumptions about the oracles and the class of distributions (functions) this applies to. The authors have not revised their submission to address these concerns during the discussion phase.

The only "accept" review (j1qK) is very brief and the questions asked ("Would it be possible to extend the result to some extent to non-convex landscapes?" on a paper entitled "Non-convex Optimization with Stochastic Zeroth Oracles") do not inspire much confidence.

**Reviewer Concerns:**

Assumptions about the "reproducible randomness" in the oracles utilized have been discussed (with aUSH), but have not been clarified appropriately in a revision.

Assumptions about the class of distributions (functions) have been discussed in the rebuttal (with WWpT, wZj3), but have not been explained appropriately in the revision.

A fairer discussion of the quantum speedup (i.e., comparison against classical algorithms) required by aUSH has not been provided.

**Reviewer Scores:**

The most knowledgeable reviewer (wZj3) suggests a weak reject and the authors did not bother to revise the submission to address their suggestions, so I would expect no change of the score.

Reviewer aUSH, who originally suggested a strong reject (2), has been promised a revision of the submission, but this has not materialized. While he suggested that he would increase the score, the AC would advise against that, given the revision of the submission did not address the concerns about the assumptions on the oracles used appropriately.

The only "accept" review (j1qK) is very brief and the questions asked ("Would it be possible to extend the result to some extent to non-convex landscapes?" on a paper entitled "Non-convex Optimization with Stochastic Zeroth Oracles") do not inspire confidence. I imagine that a further discussion may result in a less confident review.

---

### Decision · Program_Chairs · 2026-01-26

Reject